# VODCA2GPP - A new global, long-term (1988-2020) GPP dataset from microwave remote sensing

Benjamin Wild[1], Irene Teubner[1,2], Leander Moesinger[1], Ruxandra-Maria Zotta[1], Matthias Forkel[3], Robin van der Schalie[4], Stephen Sitch[5] and Wouter Dorigo[1]

[1]Department of Geodesy and Geoinformation, TU Wien, Wiedner Hauptstraße 8, 1040 Vienna, Austria
[2]Zentralanstalt für Meteorologie und Geodynamik (ZAMG), Hohe Warte 38, 1190 Vienna, Austria
[3]Environmental Remote Sensing Group, Institute of Photogrammetry and Remote Sensing, Technische Universität Dresden, Helmholtzstraße 10, 01069 Dresden, Germany
[4]VanderSat, Wilhelminastraat 43A, 2011 VK Haarlem, the Netherlands
[5]College of Life and Environmental Sciences, University of Exeter, Exeter, EX4 4QE, UK

*Correspondence to*: Benjamin Wild (benjamin.wild@tuwien.ac.at)

**Abstract.** Long-term global monitoring of terrestrial Gross Primary Production (GPP) is crucial for assessing ecosystem response to global climate change. In recent decades, great advances have been made in estimating GPP and many global GPP datasets have been published. These datasets are either based on observations from optical remote sensing, are upscaled from in situ measurements, or rely on process-based models. Although these approaches are well established within the scientific community datasets nevertheless differ significantly.

Here, we introduce the new VODCA2GPP dataset, which utilizes microwave remote sensing estimates of Vegetation Optical Depth (VOD) to estimate GPP at global scale for the period 1988 - 2020. VODCA2GPP applies a previously developed carbon sink-driven approach (Teubner et al., 2019, 2021) to estimate GPP from the Vegetation Optical Depth Climate Archive (Moesinger et al., 2020; Zotta et al., in prep.), which merges VOD observations from multiple sensors into one long-running, coherent data record. VODCA2GPP was trained and evaluated against FLUXNET in situ observations of GPP and compared against largely independent state-of-the-art GPP datasets from MODIS, FLUXCOM and the TRENDY-v7 process-based model ensemble.

The site-level evaluation with FLUXNET GPP indicates an overall robust performance of VODCA2GPP with only a small bias and good temporal agreement. The comparisons with MODIS, FLUXCOM and TRENDY-v7 show that VODCA2GPP exhibits very similar spatial patterns across all biomes but with a consistent positive bias. In terms of temporal dynamics, a high agreement was found for regions outside the humid tropics, with median correlations around 0.75. Concerning anomalies from the long-term climatology, VODCA2GPP correlates well with MODIS and TRENDY-v7 (Pearson's r: 0.53 and 0.61) but less well with FLUXCOM (Pearson's r: 0.29). A trend analysis for the period 1988-2019 did not exhibit a significant trend in VODCA2GPP at global scale but rather suggests regionally different long-term changes in GPP. For the shorter overlapping observation period (2003-2015) of VODCA2GPP, MODIS, and the TRENDY-v7 ensemble, significant increases of global

GPP were found. VODCA2GPP can complement existing GPP products and is a valuable dataset for the assessment of large-scale and long-term changes in GPP for global vegetation and carbon cycle studies. The VODCA2GPP dataset is available at TU Wien Research Data (https://doi.org/10.48436/1k7aj-bdz35; Wild et al., 2021).

## 1 Introduction

Gross Primary Production (GPP) describes vegetation's conversion of atmospheric $CO_2$ to carbohydrates through photosynthesis and is the largest $CO_2$ flux in the carbon cycle (Beer et al., 2010). GPP is also considered the primary driver of the terrestrial carbon sink responsible for the uptake of approximately 30% of anthropogenic $CO_2$ emissions (Friedlingstein et al., 2020). GPP therefore plays a key role in mitigating the negative effects of anthropogenic emissions. Estimates of global mean annual GPP range from 112 (Anav et al., 2015) up to 175 (Welp et al., 2011) Pg C yr$^{-1}$, but exhibits a high degree of interannual variability. It is strongly affected by increasing concentrations of $CO_2$ in the atmosphere and the associated global climate change (Haverd et al., 2020; Schimel et al., 2015; Cox et al., 2000). Quantifying GPP is essential to understand the effect of climate variability and changes in atmospheric $CO_2$ concentrations on the land carbon cycle (e.g., Baldocchi et al., 2016; Nemani et al., 2003).

Locally, GPP can be determined at in-situ flux towers, which measure the net exchange of carbon dioxide by means of eddy-covariances that are partitioned into GPP and ecosystem respiration fluxes (Baldocchi, 2003). FLUXNET (Pastorello et al., 2020) is the global network of flux towers covering all major biomes, and provides the scientific community with harmonized and well-documented flux observations. FLUXNET stations, however, are sparsely and unevenly distributed, which complicates the derivation of GPP globally.

On the global scale, GPP is commonly estimated using optical remote sensing data in combination with (semi-)empirical or machine learning models (e.g., O'Sullivan et al., 2020; Jung et al., 2020; Gilabert et al., 2017; Alemohammad et al., 2017; Tramontana et al., 2016). Specifically, these models are based on light use efficiency (LUE) theory and/or statistical models that are applied to derive GPP based on optical remote sensing variables that are indicative of the vegetation's photosynthetic activity, such as the Fraction of Absorbed Photosynthetically Active Radiation (fAPAR), Leaf Area Index (LAI), spectral vegetation indices, or Sun Induced Fluorescence (SIF). Optical remote sensing-based datasets have the advantage of being available globally with high spatial (usually on the order of 100 m to 1 km) and temporal resolution (e.g., 8-daily for MODIS). However, optical remote sensing is strongly affected by cloud cover, leading to data gaps and high uncertainties in regions with frequent cloud cover and high GPP such as tropical forests. Additionally, in very productive regions methods based on optical remote sensing tend to underestimate GPP because of the saturation of reflectance measurements in dense canopies (Turner et al., 2006).

Compared to optical remote sensing, Vegetation Optical Depth (VOD) from microwave remote sensing is much less affected by weather conditions. VOD describes the vegetation's attenuation of radiation in the microwave domain, which is controlled by its water content, biomass, type, and density (Jackson and Schmugge, 1991; Vreugdenhil et al., 2016). Thus, VOD has been intensively used as a proxy of above-ground biomass (Li et al., 2021; Rodríguez-Fernández et al., 2018; Tian et al., 2016; Liu et al., 2015) and is becoming increasingly important for monitoring vegetation dynamics (e.g., Frappart et al., 2020; Piles et al., 2017).

Teubner et al. (2018, 2019, 2021) investigated how GPP can be estimated from VOD. First, they showed that GPP is significantly correlated with spatial patterns and temporal changes in VOD (Teubner et al., 2018). Based on this relationship, they developed a theoretical framework and a machine-learning method using FLUXNET observations to predict GPP using VOD (Teubner et al., 2019). They showed that GPP can be adequately estimated for most regions of the world with an overall tendency for moderate overestimation and good temporal agreement with existing GPP products, especially for temperate regions. Recently, Teubner et al. (2021) improved this method by adding air-temperature in their model to account for temperature dependence of plant respiration (Atkin and Tjoelker, 2003), and found that this significantly improved the temporal agreement with reference GPP data.

However, until recently, long-term analysis of GPP from VOD was complicated due to relatively short observation periods of individual passive microwave remote sensing sensors (Moesinger et al., 2020). Moesinger et al. (2020) overcame this issue by merging single-frequency VOD from various sensors into the long-term Vegetation Optical Depth Climate Archive (VODCA), which comprises VOD observations of more than 20 years for X-, and C-Band and more than 30 years for Ku-Band. A new version of VODCA (Zotta et al., in prep.) not only combines single sensors from identical frequencies but also merges observations from different bands (X, C and Ku) into a single long-running, multi-frequency VOD climate archive with improved quality.

Here, our objective is to generate, evaluate and describe a novel long-term GPP dataset by applying the approach of Teubner et al. (2019, 2021) to the VODCA dataset. This microwave-based GPP dataset can likely complement existing datasets from optical satellite observations as it is less affected by cloud cover which enables a consistent long-term analysis of changes in global GPP. In our analysis, we compare the VODCA2GPP dataset mainly with other data-driven products (FLUXNET, MODIS and FLUXCOM). However, FLUXCOM does not account for $CO_2$ fertilization effects (Walker et al, 2020) which is why trends derived from FLUXCOM are not realistic (Jung et al., 2020). Therefore, we assess monthly anomalies and long-term trends in VODCA2GPP also against TRENDY models, which consider $CO_2$ fertilization.

## 2 Data

## 2.1 Input to the VODCA2GPP model

**2.1.1 VODCA**

The Vegetation Optical Depth Climate Archive (VODCA v1; Moesinger et al., 2020) consists of three single-frequency VOD products (Ku-, X- and C-band), covering the period from 1987-2017 (Ku-Band), 1997-2019 (X-Band), and 2002-2019 (C-band), respectively. For VODCA2GPP, we used an updated VODCA version (VODCA v2 CXKu; Zotta et al., in prep) that merges all bands in a single dataset to obtain increased spatial and temporal coverage and reduced random errors compared to

105 VODCA v1. VODCA v2 CXKu utilizes observations from the same sensors and frequencies as VODCA v1 (Table 1) to generate a single long-running multi-frequency VOD time series. VODCA v2 CXKu is obtained by first scaling VODCA v2 observations from C- and Ku-band to X-band to remove systematic biases and then computing a weighted average in order to fuse overlapping observations. The reference frequency for the scaling of the different frequencies is X-Band. VODCA v2 CXKu provides a single, long-term vegetation metric covering over 30 years of observations (1988-2020) and thus exceeds

the temporal length of the single-frequency multi-sensor products (VODCA v2 C-, X and Ku). VODCA v2 CXKu merges 15 passive night-time VOD datasets retrieved from seven different sensors via the Land Parameter Retrieval Model (LPRM; Van der Schalie et al., 2017). LPRM is based on radiative transfer theory introduced by Mo et al. (1982) and uses forward modelling to simulate the top of atmosphere brightness temperatures under a wide range of conditions and minimizes its difference with the actual satellite observation. Although primarily developed for soil moisture, it simultaneously solves for the VOD using

an analytical solution by Meesters et al. (2005), utilizing the ratio between H- and V-polarized observations (Van der Schalie et al., 2017). LPRM assumes that the soil and vegetation temperatures are equal, which may not be the case during the day due to uneven heating from solar radiation. VODCA v2 therefore uses only night-time observations which are assumed to be in thermal equilibrium (Owe et al., 2008). Scaling of the single-sensor VOD observations is done by means of cumulative distribution function (CDF) matching (Moesinger et al., 2020).

**Table 1: Input data for the merged-band VODCA v2 with the main sensor specifications: time periods used, local ascending equatorial crossing times (AECT) and used frequencies. Table information is taken from Moesinger et al. (2020) and adapted for VODCA v2 CXKu.**

| Sensor | Time period used | AECT | C-Band [GHz] | X-Band [GHz] | Ku-Band [GHz] | reference |
|---|---|---|---|---|---|---|
| AMSR-E | Jun 2002 – Oct 2011 | 13:30 | 6.93 | 10.65 | 18.70 | Van der Schalie et al. (2017) |
| AMSR2 | Jul 2012 – Dec 2020 Jul 2012 – Aug 2017 (Ku-Band) | 13:30 | 6.93, 7.30 | 10.65 | 18.70 | Van der Schalie et al. (2017) |
| SSM/I F08 | Jul 1987 – Dec 1991 | 18:15 | | | 19.35 | Owe et al. (2008) |
| SSM/I F011 | Dec 1991 – May 1995 | 17:00-18:15 | | | 19.35 | Owe et al. (2008) |
| SSM/I F13 | May 1995 – Apr 2009 | 17:45 – 18:40 | | | 19.35 | Owe et al. (2008) |
| TMI | Dec 1997 – Apr 2015 | Asynchronous | | 10.65 | 19.35 | Owe et al. (2008); Van der Schalie et al. (2017) |
| WindSat | Feb 2003 – Jul 2012 | 18:00 | 6.80 | 10.70 | 18.70 | Owe et al. (2008); Van der Schalie et al. (2017) |

The preprocessing of LPRM level 2 VOD data used in VODCA v2 follows the steps described in detail in Moesinger et al.
(2020). These include projecting the data onto a $0.25° \times 0.25°$ grid, using nearest neighbour resampling, selecting the closest
night-time value in a window of $\pm$ 12 hours for every 0:00 UTC (Zotta et al., in prep.). Data are masked for radio-frequency
interference (De Nijs et al., 2015), negative VOD retrievals and temperatures. Different from Moesinger et al. (2020), masking
for low land surface temperature (LST < 0°C), when the dielectric properties of water change drastically, is not based on Ka-
band retrievals because these have high uncertainties over frozen land (Holmes et al. 2009). Instead, VODCA v2 uses the
ERA-5 Land (Muñoz-Sabater et al. 2021) soil temperature level 1 (stl1) data. To ensure that all observations taken under frozen
conditions are masked, all observations with an associated surface soil temperature (stl1) below 3°C are masked (Zotta et al.,
in prep.).

### 2.1.2 ERA5-Land – 2m Air Temperature

2 m air temperature (T2m) from the ERA5-Land dataset was used to represent the temperature dependence of autotrophic
respiration. T2m is a commonly used parameter for describing the relationship between autotrophic respiration and temperature
(Teubner et al., 2021; Drake et al., 2016; Ryan et al., 1997). ERA5-Land is a reanalysis dataset of meteorological variables
which is provided by the European Centre for Medium-Range Weather Forecasts (ECMWF) (Muñoz-Sabater et al., 2021).
ERA5-Land is produced at a spatial resolution of 9 km (~0.08°) and is available hourly.

### 2.1.3 FLUXNET2015 in-situ GPP

In-situ GPP data from Tier1 v1 FLUXNET2015 (Pastorello et al., 2020) were used to train and evaluate the VODCA2GPP
product. FLUXNET GPP estimates are available for night-time and day-time flux partitioning, which were averaged as
suggested by Pastorello et al. (2020). FLUXNET data are available daily from 1991 until 2014 with a mean observation
timespan of $7.27 \pm 4.89$ years for the used stations, indicating significant variability in station data availability. An overview
of the used FLUXNET2015 stations can be found in Table B1.

### 2.2 Reference datasets

### 2.2.1 MODIS GPP

GPP estimates derived from Moderate Resolution Imaging Spectroradiometer (MODIS) satellite data are based on Monteith's
(1972) light-use efficiency concept which relates the amount of absorbed solar radiation to vegetation productivity. The
MODIS algorithm uses fAPAR as proxy for the absorbed solar energy. For this study the MOD17A2H v006 GPP product was
used (Running et al., 2015; Zhao et al., 2005). It is available at 8-daily temporal resolution and 500 m sampling and was
resampled to 0.25° to match the resolution of VODCA2GPP.

### 2.2.2 FLUXCOM GPP

FLUXCOM GPP (Tramontana et al., 2016; Jung et al., 2020) is produced by upscaling GPP estimates from in-situ eddy covariances using machine learning techniques. Two FLUXCOM GPP setups exist: FLUXCOM RS uses high resolution land surface properties from MODIS observations as machine learning model input while FLUXCOM RS + METEO uses the mean seasonal cycle of land surface variables derived from MODIS observations and additionally incorporates meteorological data (Jung et al., 2020). For validating VODCA2GPP, FLUXCOM RS was used because it includes temporal properties of land surface variables at finer spatial and temporal resolution than FLUXCOM RS+METEO. FLUXCOM RS GPP has 10 km sampling and is available every 8 days in accordance with the MODIS input data. The data were aggregated to 0.25° to match the VODCA2GPP resolution.

### 2.2.3 TRENDY-v7 GPP

In addition to remote sensing-based datasets, GPP estimates from the reanalysis-driven TRENDY-v7 ensemble of 16 dynamic global vegetation models (DGVMs) were used as an independent reference dataset (Le Quéré et al. 2018; Sitch et al. 2015). TRENDY-v7 simulations consider forcing effects of climate, land use, and changes in atmospheric $CO_2$ concentration on GPP over the period 1950-2017. The TRENDY-v7 ensemble consists of the following DGVMs: CABLE-POP, CLASS-CTEM, CLM5.0, DLEM, ISAM, JSBACH, JULES, LPJ, LPJ-GUESS, LPX, OCN, ORCHIDEE, ORCHIDEE-CNP, SDGVM, SURFEX and VISIT. DGVMs output monthly GPP which was provided on a common $1° \times 1°$grid. For the comparison with VODCA2GPP, all TRENDY-v7 models were regridded to 0.25° using nearest neighbour resampling and merged into an unweighted ensemble mean GPP time series.

## 3 Methods

### 3.1 VOD2GPP-model

VODCA2GPP is based on the VOD-driven GPP estimation approach (the VOD2GPP-model) introduced by Teubner et al. (2019, 2021). The VOD2GPP model describes the theoretical relationship between GPP and VOD. The biogeochemical basis of this model is the relationship between GPP and ecosystem net uptake of carbon (NPP) and autotrophic respiration ($R_a$) (Bonan, 2008):

$$GPP = R_a + NPP, \tag{3.1}$$

where Ra can be again split into two terms: maintenance respiration and growth respiration (Bonan, 2008). The VOD2GPP-model makes use of several VOD variables to represent the sum of NPP and Ra: the original VOD time series (VOD), which relates to maintenance respiration, temporal changes in VOD ($\Delta$(VOD)), which relate to both growth respiration and NPP, and

the temporal median of VOD (mdn(VOD)) derived from the complete time series which serves as a proxy of vegetation density. Specifically, mdn(VOD) is incorporated to subtract larger structural vegetation components which makes the resulting model more closely related to biomass changes of smaller structural vegetation components such as leaves. It was shown that this increases model performance (Teubner et al., 2019). NPP is mostly represented by Δ(VOD) while Ra is represented by the original VOD signal and Δ(VOD). Thus, the VOD-based-only VOD2GPP-model can be formulated as follows (Teubner et al., 2019):

$$GPP(VOD) = s(VOD) + s(\Delta(VOD)) + s(mdn(VOD)),$$ (3.2)

where $s()$ denotes the mapping function that maps the input variables to GPP. Δ(VOD) is derived for each pixel ($x_i$) by computing the difference between two consecutive VOD observations of the smoothed and 8-daily aggregated VOD Signal (Teubner et al. 2019):

$$\Delta(VOD) = VOD_{x_i,t_j} - VOD_{x_i,t_{j-1}}$$

The smoothing was performed in order to increase the robustness of the derivation and implemented using a Savitzky-Golay filter with a window size of 11 data points as suggested by Teubner et al. (2021).

Eq. 3.2 represents a simplified model formulation connecting VOD to GPP but which does not explicitly take into account the strong temperature dependence of autotrophic respiration (Wythers et al., 2013; Atkin et al., 2005; Tjoelker et al., 2001) which is mainly attributed to its maintenance part (Bonan, 2008; Ryan et al., 1997). Therefore, an improved formulation of the model was developed by considering the temperature dependence of maintenance respiration through a term representing the interaction between temperature (T2m) and VOD (Teubner et al., 2021):

$$GPP(VOD, T2m) = te(VOD, T2m) + s(\Delta(VOD)) + s(mdnVOD),$$ (3.3)

The mapping and interaction functions were implemented using generalized additive models (GAMs). The usage of short time intervals (in the order of several days) for the computation of Δ(VOD) is crucial since it reduces the influence of larger vegetation components (e.g., stems) and makes the model more sensitive to changes in leaf biomass.

## 3.2 Generalized Additive Models

Generalized Additive Models (GAMs; Hastie and Tibshirani, 1990) are semi-parametric generalizations of linear models and combine properties of Generalized Linear Models (GLM) and additive models (Guisan et al., 2002). Link functions $f()$ are trained and summed up for each predictor in order to relate the expected value of a response variable $E(Y)$ to the explanatory variables $x_i$ (Hastie and Tibshirani, 1990). The model can be written as:

$$215 \quad E(Y) = \beta + \sum_{i=1}^{n} s_i(x_i), \tag{3.4}$$

where $\beta$ denotes a constant offset and $n$ corresponds to the number of input predictor variables $x_i$. The link functions $s_i()$ are implemented as smooth spline functions and allow the representation of non-monotonic and non-linear relationships which give them a high degree of flexibility (Hastie and Tibshirani, 1990). Hence, the relationship between target and predictor

variables does not require explicit a-priori knowledge but can be estimated from the data itself, which makes GAMs appropriate for the VOD2GPP-model for which the exact relationship between VOD, air temperature, and GPP is difficult to determine (Teubner et al., 2019).

### 3.3 Preprocessing

The model-input data (response variable: FLUXNET GPP; predictor variables: VODCA v2 CXKu, ERA5-Land T2m) were

225 resampled from daily to 8-daily resolution using the 8-day means over the respective time period in order to reduce noise and computation times. This means that also the final VODCA2GPP represents the mean of daily GPP for an 8-day period with an estimate every 8 days. Since VODCA v2 CXKu already incorporates extensive quality flagging (e.g., for temperature) no additional data processing was necessary.

### 3.4 Model training and output

For each valid FLUXNET2015 in-situ observation, the corresponding overlapping pixel values of VOD, Δ(VOD), mdn(VOD) and T2m were used to set up the GAM. Data from days with one or more invalid or missing observations were not considered for model training. While a cross validation was performed to evaluate the model (Sect. 3.5), all data were used for training of the final VODCA2GPP model. The GAM-based implementation of the VOD2GPP-model is consistent with Teubner et al. (2021) and utilizes algorithms from the pygam python package (Servén et al., 2018).

The trained VODCA2GPP model was applied to each pixel where all input variables (VOD, Δ(VOD), mdn(VOD) and T2m) were available. The result of this upscaling process is VODCA2GPP which covers the period between January 1988 and July 2020. It has a spatial resolution of 0.25° and its temporal resolution is 8 days. In rare cases (~2.5% of all data points) the machine learning model produced slightly negative results for GPP, due to the extrapolation capacities of the trained GAM.

As negative GPP is not possible, such estimates were set to zero.

### 3.5 Site-level evaluation and uncertainty assessment

The robustness of the model was evaluated based on a site-based cross-validation analysis during which the influence of the selection of available in-situ stations on the GPP model was investigated. For the cross-validation 10 VODCA2GPP models were trained. Each of the 10 models was trained with 90% of the available FLUXNET stations, while the remaining 10% of

the stations were retained for validation (Teubner et al., 2019). Every station was excluded exactly once which is why this approach is classified as pseudo-random. Model performance was assessed at all sites that were omitted in the respective model run by computing Root Mean Square Errors (RMSE), Bias, and Pearson's r for different time scales (8-daily, monthly and yearly). In addition, the uncertainty of the VODCA2GPP model was then expressed through the minimum/maximum range as well as the standard deviation of the resulting 10 mean annual accumulated GPP estimates for each pixel. The standard deviation is also incorporated as an uncertainty map in the available dataset (layer name: 'Uncertainties') to support users with an indicator for known uncertainties in VODCA2GPP.

### 3.6 Product evaluation and assessment

Mean annual, monthly, and 8-daily GPP from VODCA2GPP, MODIS GPP and FLUXCOM GPP were evaluated against GPP from FLUXNET. The used error metrics were RMSE, Bias and Pearson's r. Global spatial GPP patterns were compared against products by computing the mean annual GPP per pixel and the differences in mean annual GPP per pixel over the common observation period. Temporal agreements were tested by means of a Pearson correlation analysis for 8-daily GPP. A correlation analysis of GPP anomalies was conducted for monthly GPP and also includes TRENDY-v7 GPP which only provided monthly GPP data. Anomalies were derived by subtracting the long-term mean of the overlapping observation periods from monthly GPP estimates for each product.

Additionally, a trend analysis was conducted for all available GPP products in order to compare long-term changes in GPP. Trends in yearly median GPP were quantified using the Theil-Sen estimator (Theil, 1950; Sen, 1968) which calculates the slopes for each line between two points. The median of all computed slopes is then used for line-fitting making it insensitive to outliers and more robust than simple linear regression (Wilcox, 2010). Slopes were considered as significant when the signs of the lower and upper 90%-confidence intervals were equal. For the trend analysis yearly median GPP was used.

# 4 Results

## 4.1 Spatiotemporal patterns in global GPP

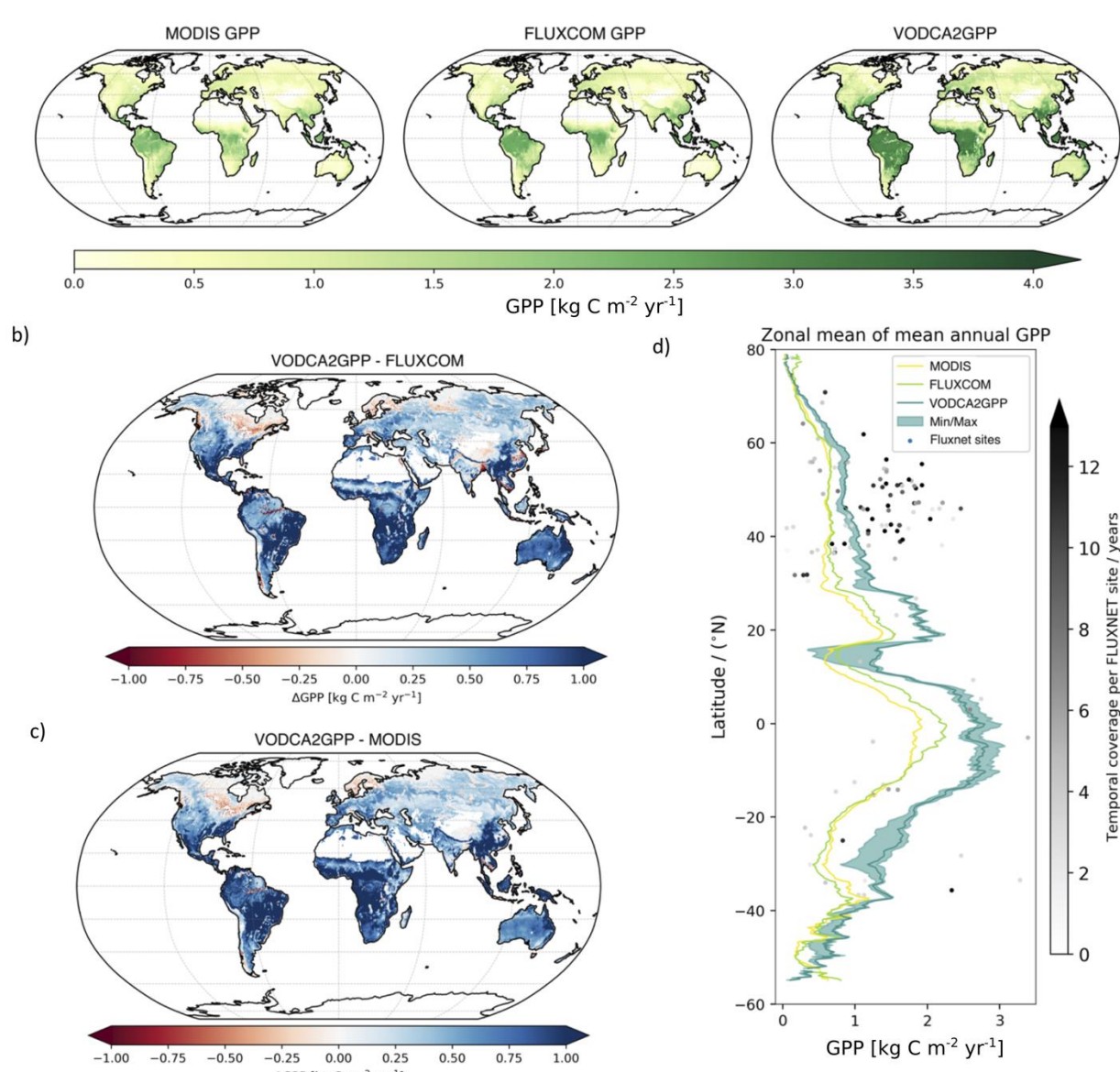

**Figure 1: a) Mean yearly aggregated GPP for the common observation period of the three products VODCA2GPP, MODIS, and FLUXCOM (2002-2016); b) and c) Difference in mean annual GPP between VODCA2GPP and FLUXCOM and MODIS, respectively; d) Latitude plot of zonal means of mean annual GPP. The means were computed based on 8-daily, 0.25 degree sampling. The Min/Max area denotes the minimum/maximum latitudinal mean for the ten model runs that were obtained with the site-based cross-validation. The dots represent the latitudinal location of the FLUXNET sites and their corresponding mean annual GPP. The**

brightness of the dots indicates the data availability for the respective FLUXNET station. Only pixels and observation dates that are
available in all three datasets were used for these plots.

The average annual GPP of VODCA2GPP exhibits spatial patterns similar to the remote sensing-based MODIS and FLUXCOM products (Fig. 1). The agreement in annual GPP is high in northern latitudes (e.g., Europe, Russia, Canada) while there are relatively large differences in the southern hemisphere, especially in tropical and sub-tropical regions (Fig. 1d). The largest positive differences are found in the subtropics. Very arid regions (e.g., Australian deserts, Kalahari Desert, etc.) have
280 low mean yearly productivity in all three datasets (Fig. 1a) but tend to be higher in VODCA2GPP compared to MODIS and FLUXCOM (Fig. 1b, c). The mean global total GPP as derived from VODCA2GPP amounts to $200 \pm 2.2$ Pg C yr$^{-1}$. Comparison of the latitudinal distribution of FLUXNET stations shows that closest agreement in yearly GPP is generally found in regions with high density of FLUXNET in-situ stations while largest discrepancies are found in regions with few or no FLUXNET stations.

Similarly, uncertainties tend to be smaller in latitudes with a high density of FLUXNET stations (Fig. 1d). The lowest spread in the 10 models (i.e., the lowest uncertainty) is found north of 20°N where also the majority of FLUXNET GPP stations is located. The Southern hemisphere, where only few in-situ stations are located, generally exhibits a larger spread (higher uncertainty) indicating a considerable sensitivity of the model to the choice of stations. This emphasizes the need for a well
distributed network of in-situ flux towers across all biomes. The uncertainty map (Fig. 2b) shows that arid regions (e.g., Sahara, Australian deserts, Arabian Peninsula) as well as various mountainous regions (e.g., Carpathians, Alps, Rocky Mountains, Andes) have the highest model uncertainties. Moderate to high model uncertainties are also exhibited for the tropics. Furthermore, significant uncertainties in VODCA2GPP are found in parts of eastern and western Siberia's boreal forests as well as in parts of southern China.

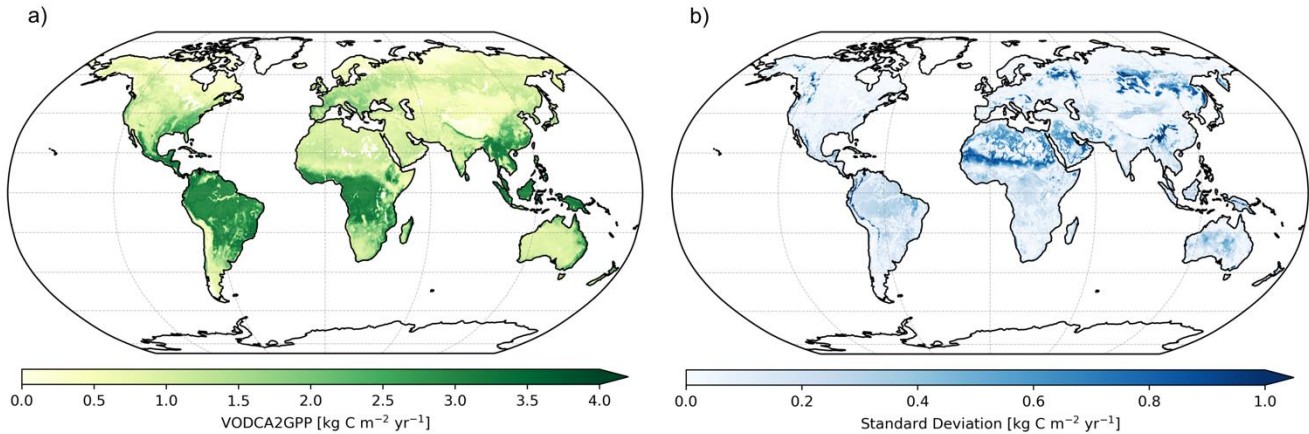

**Figure 2: a) Mean annual GPP as derived from VODCA2GPP for the period 1988-2019. b) Standard deviation of mean annual GPP (1988-2019) as obtained by the uncertainty analysis based on site-level cross-validation.**

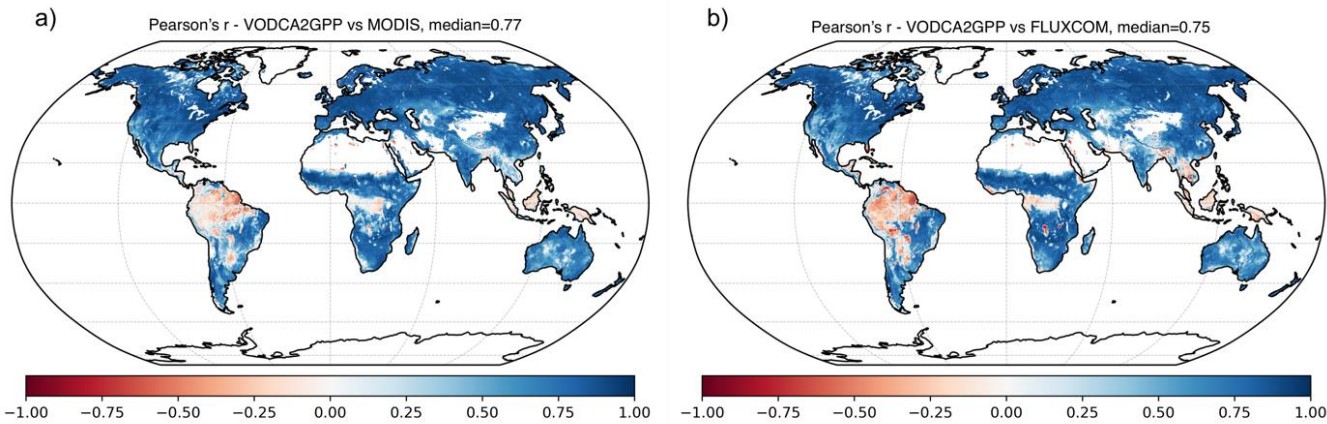

**Figure 3: Pearson's r between VODCA2GPP and MODIS GPP (a) and VODCA2GPP and FLUXCOM GPP (b). The correlations are based on the common observation period between 2002 and 2016 with 0.25° spatial and 8-daily temporal resolution.**

VODCA2GPP shows good temporal agreement with MODIS and FLUXCOM for most regions outside the tropics (Fig. 3). Pearson's r is the highest in regions with distinct interannual variability such as sub-arctic, temperate, and semi-arid regions and the lowest for dense tropical forests where even negative correlations occur. Median Pearson's r between VODCA2GPP and the reference datasets MODIS and FLUXCOM is 0.77 and 0.75, respectively.

**4.2 Site-level evaluation**

VODCA2GPP's tendency towards a positive bias with respect to MODIS and FLUXCOM products is not mirrored in the comparison against FLUXNET GPP (Fig. 4, Fig. A1). The bias with respect to FLUXNET site data is substantially smaller for VODCA2GPP than for MODIS and FLUXCOM. The RMSE and Pearson's r values of VODCA2GPP are slightly higher and lower, respectively, than for MODIS and FLUXCOM and of the same magnitude for 8-daily values and mean annual GPP. All three datasets underestimate productivity at high GPP values. A landcover-based analysis (Fig. A3) shows that discrepancies in annual VODCA2GPP are mostly occurring in (semi-)arid environments (e.g., savannas, open shrublands, grasslands). VODCA2GPP performs best in temperate environments (e.g., wetlands, evergreen broadleaf forest, croplands). Wetlands and evergreen broadleaf forests exhibit the best performance for all products while all three datasets underperform in open shrublands and deciduous broadleaf forest.

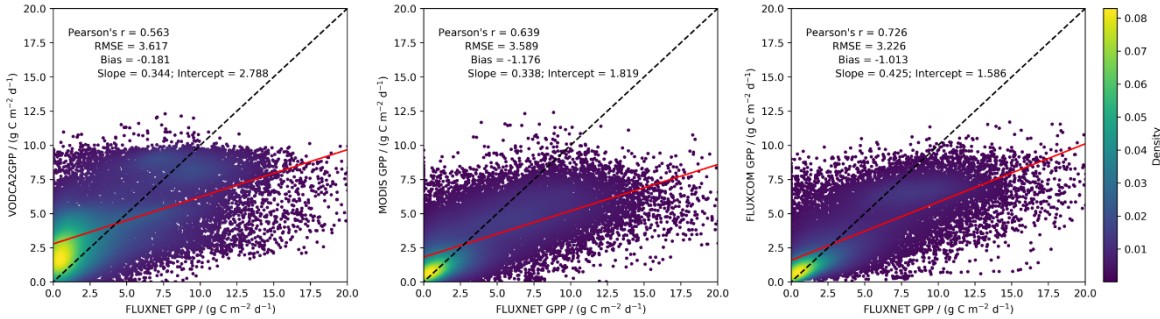

**Figure 4: GPP from FLUXNET plotted against GPP from VODCA2GPP, MODIS and FLUXCOM for the period 2002-2016 with 8-daily sampling.**

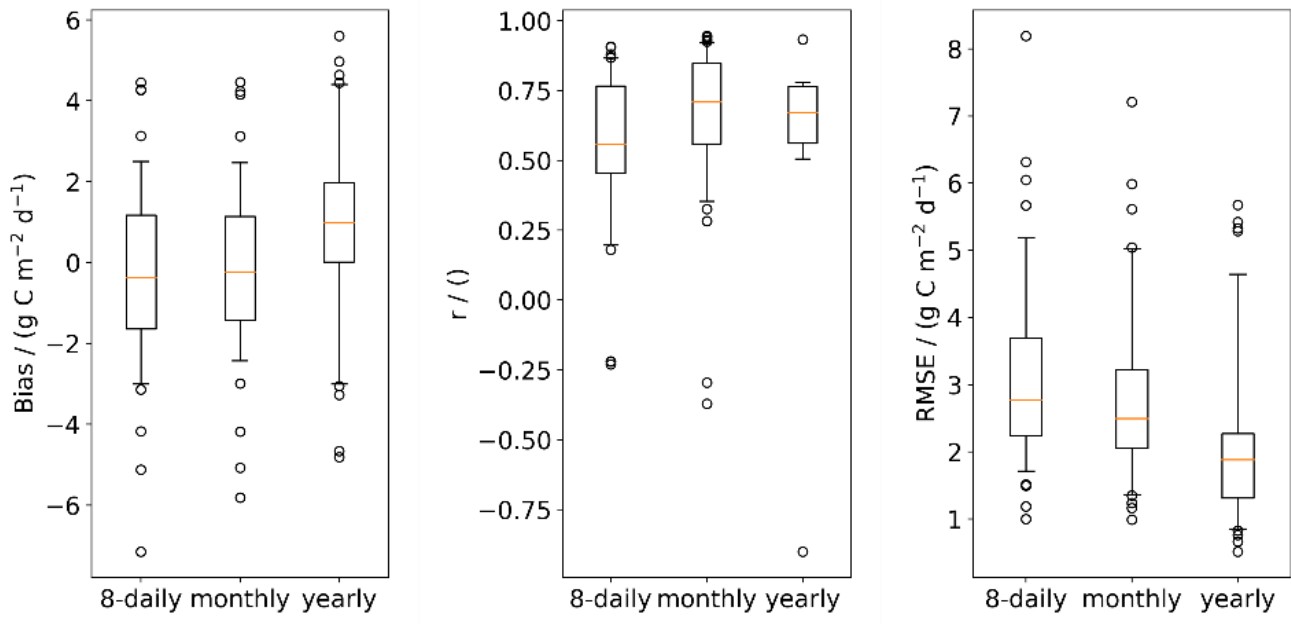

**Figure 5: Site-based cross-validation for 8-daily, monthly, and yearly sampling of GPP from VODCA2GPP and FLUXNET. RMSE, Bias and Pearson's r were computed at each of the 10% of FLUXNET sites that were omitted during the respective training run. Non-significant Pearson correlation (p-value < 0.1) were ignored. The boxplots for Pearson's r are based on the 71 (8-daily), 66 (monthly) and 8 (yearly sampling) significant values for Pearson's r values. The whiskers of the boxplots extend to the 0.05/0.95 percentiles.**

The site-based spatial cross-validation also exhibits only a small (negative) bias of VODCA2GPP (Fig. 5) for monthly and 8-daily GPP values while the bias for annual variations is positive and slightly higher. High median Pearson's r for 8-daily and monthly values indicates good model performance for seasonal variations. It is to be noted that there are only 8 significant Pearson's r values for yearly sampling which decreases the expressiveness of this value. This is explicable with the short

observation period of most FLUXNET sites which might not exhibit interannual variability. The RMSE decreases with
increasing observation length scales.

## 4.3 Anomaly patterns in space and time

In terms of anomalies from the long-term climatology, VODCA2GPP shows good correlation with MODIS and TRENDY-v7 and weaker correlation with FLUXCOM (Table 2). TRENDY-v7 correlates similarly well with VODCA2GPP and MODIS and shows worse correspondence with FLUXCOM. The highest correlation is found between the two optical remote sensing-based products MODIS and FLUXCOM.

**Table 2: Pearson's r correlation matrix for mean global monthly GPP anomalies between 2002 and 2016.**

| | *VODCA2GPP* | *MODIS GPP* | *FLUXCOM GPP* | *TRENDY-v7 GPP* |
|---|---|---|---|---|
| *VODCA2GPP* | 1.00 | | | |
| *MODIS GPP* | 0.53 | 1.00 | | |
| *FLUXCOM GPP* | 0.29 | 0.69 | 1.00 | |
| *TRENDY-v7 GPP* | 0.61 | 0.60 | 0.26 | 1.00 |

The temporal evolution and spatial distribution of the anomalies exhibit similar patterns (Fig. A4). Several extreme events are captured in VODCA2GPP and in at least one of the other GPP datasets. An example of such GPP extremes are the strong positive anomalies between 2010 and 2011 at around 25°S which were mainly caused by record-breaking rainfalls in Australia (Wardle et al., 2013). These positive anomalies are clearly visible in all examined GPP products apart from FLUXCOM (Fig. A4). Other GPP extremes that are noticeable in all products apart from FLUXCOM are the extremely low GPP in 2002/2003 and early 2005 around 20°S (Fig. A4). Both anomalies can be explained by extreme drought events that occurred in these years (Bureau of Meteorology, 2002/2003/2005; Horridge et al., 2005) which are associated with El Niño events (Taschetto and England, 2009). Also, the distinct drop in GPP in 2015/2016 in similar latitudes is likely linked to El Niño related drought events (Malhi et al., 2018; Zhai et al., 2016). Generally, extreme events in VODCA2GPP are more pronounced than in the other datasets.

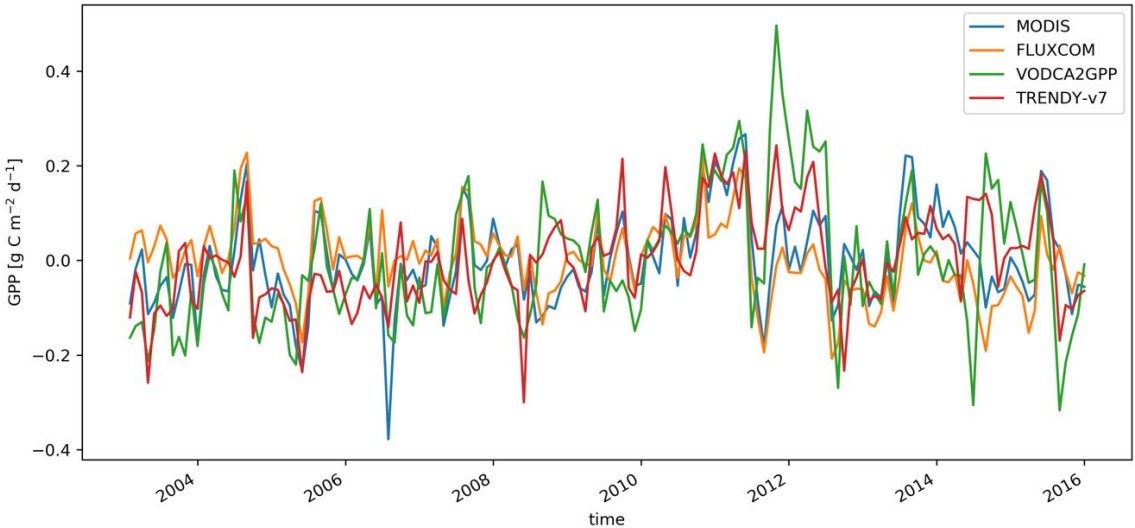

 **Figure 6: Time-series of mean global monthly GPP anomalies.**

**4.4 Global GPP trends**

Trends in global annual median GPP for the overlapping period between 2003 and 2015 are similar for VODCA2GPP, TRENDY-v7 and MODIS and all show significant positive trends (Table 3, Fig. 7). FLUXCOM does not exhibit a significant trend. The spatial distribution of GPP trends for the period 2003-2015 (Fig. 8) exhibits many similarities between all analysed 355 products. Large patterns of strong positive trends are, for example, found in eastern parts of Siberia and China as well as in India and North America. Patterns of negative trends are found north of the Caspian Sea in all datasets. The remote sensing-based products exhibit distinct patterns of declining GPP in central Siberia and significantly increasing GPP in Western Russia. Generally, the trends of VODCA2GPP match better with MODIS and TRENDY-v7 than with FLUXCOM. While there are many similar patterns in the Northern Hemisphere, trends in the Southern Hemisphere do not match well or are even 360 contradictory. Especially in the Tropics, hardly any similarities are apparent. Note that the analysed time period is short and may be impacted by individual extreme events.

For the full time period (1988-2019), VODCA2GPP increases slightly on a global scale (Table 3), but this cannot be classified as significant due to contradictory upper and lower confidence intervals. The same is true for the slightly shorter period between 365 1988 and 2016 during which TRENDY-v7 does detect a small significant positive trend on a global scale. The spatial distribution of long-term (1988-2019; Fig. 9) trends in VODCA2GPP is similar to the shorter period (2003-2015), but in general, long-term VODCA2GPP trends are less pronounced. The comparison of the fully overlapping period between VODCA2GPP and TRENDY-v7 (1988-2016, Fig. A5) shows that TRENDY-v7 exhibits weak but consistent positive trends for practically all biomes while VODCA2GPP trends are spatially differing and for some regions even opposite in sign to the 370 trends in TRENDY-v7.

A comprehensive comparison with in-situ GPP trends is not possible because most FLUXNET time series are too short to derive reliable trends. However, trends that could be derived for a few stations with a long time series (Fig. A6) also suggest increasing GPP. The in-situ analysis indicates that there is a comparatively good correspondence between VODCA2GPP and

375 FLUXNET GPP trends. Together with the strong similarities between VODCA2GPP and MODIS/TRENDY GPP, this suggests that VODCA2GPP can provide a valuable contribution to the analysis of global GPP trends.

**Table 3: Theil-Sen trends in global yearly median GPP. Same signs of the upper/lower 90%-confidence interval indicate significant trends. The analysed periods are 2003-2015 which corresponds to the fully overlapping periods for all datasets (for the period 2002-2016 there were some data gaps in the MODIS and FLUXCOM data used at the very beginning/very end. Since these data gaps**
**could potentially impact the slope estimation, the slightly shorter period 2003-2015 was used), 1988-2016 which corresponds to the fully overlapping period of VODCA2GPP and TRENDY-v7, and 1988-2019 which corresponds to all available complete years of VODCA2GPP data.**

| | 2003-2015 | | 1988-2016 | | 1988-2019 | |
|---|---|---|---|---|---|---|
| | Theil-Sen slope [g C m-2 yr-1 ] | Lower/Upper confidence interval | Theil-Sen slope [g C m-2 yr-1 ] | Lower/Upper confidence interval | Theil-Sen slope [g C m-2 yr-1 ] | Lower/Upper confidence interval |
| *VODCA2GPP* | 0.013 | +0.008 / +0.025 | 0.002 | -0.001 / +0.006 | 0.002 | -0.001 / +0.005 |
| *TRENDY-v7 GPP* | 0.017 | +0.006 / +0.026 | 0.004 | +0.000 / +0.008 | - | - |
| *MODIS GPP* | 0.012 | +0.002 / +0.020 | - | - | - | - |
| *FLUXCOM GPP* | -0.004 | -0.009 / +0.001 | - | - | - | - |

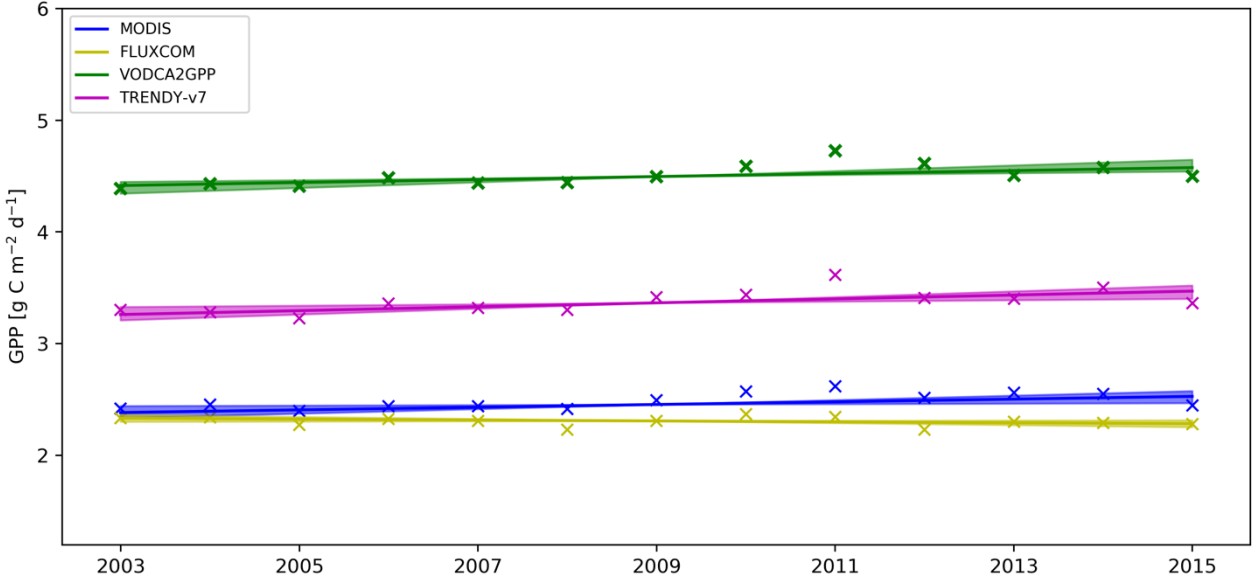

**Figure 7: Time-series of yearly median GPP with the regression lines as obtained by the Theil-Sen estimator. Areas around the regression lines indicate the 90%-confidence intervals.**

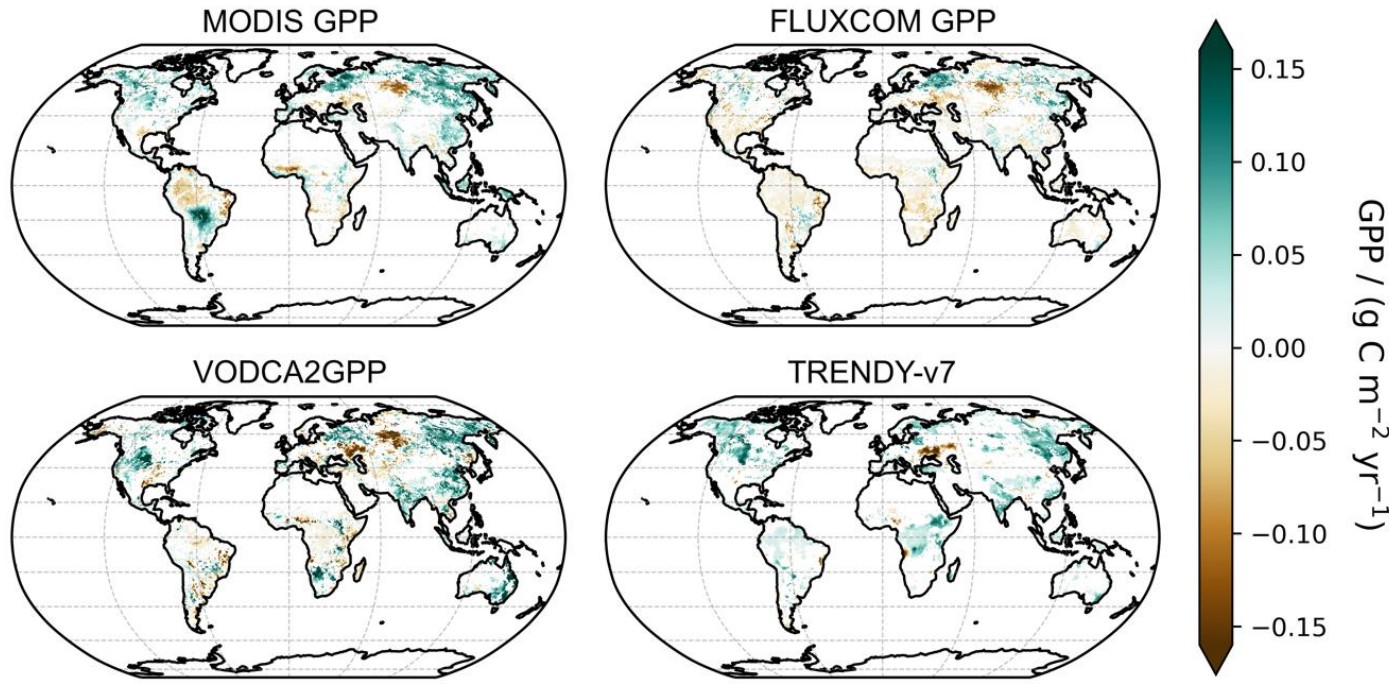

**Figure 8: Global map of trends in yearly median GPP for the period 2003-2015 for all analysed datasets. White indicates non-significant trends.**

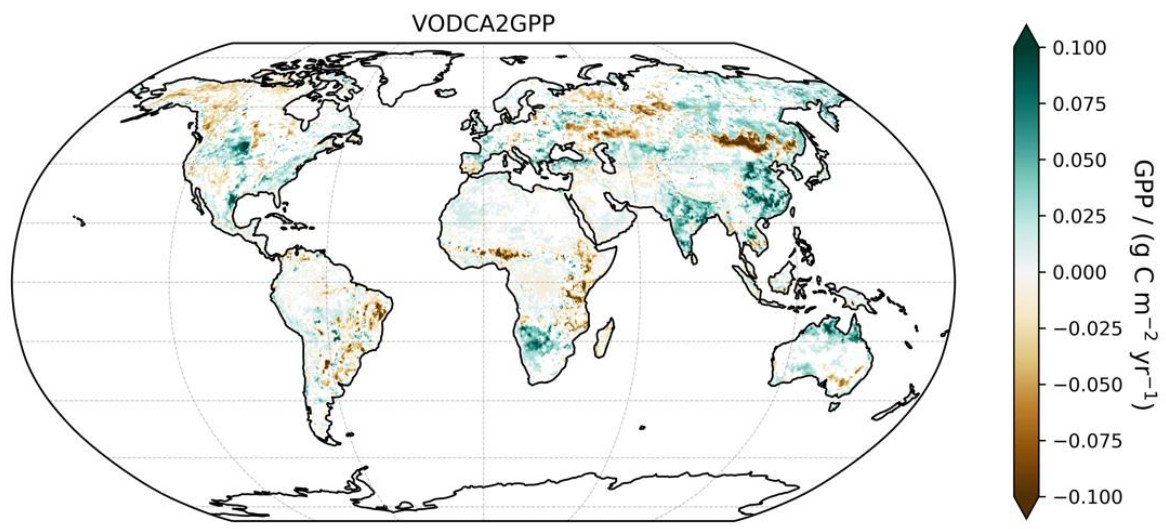

**Figure 9: Global map of yearly median GPP trends for the period 1988-2019 for VODCA2GPP. White indicates non-significant trends.**

## 5 Discussion

### 5.1 Uncertainties in the VODCA2GPP model

The results from the uncertainty analysis and the comparison with in-situ GPP show that VODCA2GPP estimates can be viewed as reliable across most biomes. However, significant uncertainties were exhibited in some areas with extreme climatic or topographic conditions (e.g., deserts and mountain ranges). Also, parts of eastern and western Siberia and parts of southern China show relatively large spread in predictions. The observed uncertainty patterns in Siberia might be associated with topography, landcover, generally lower data availability (due to frozen masking in VODCA) and a lack of FLUXNET stations. Complex topography is presumably also the main driver for uncertainties in southern China. The uncertainty analysis suggests that VODCA2GPP estimates tend to be too high in these regions and thus should be interpreted with caution. Furthermore, moderate uncertainties were also found for the tropics which is likely due to the extremely low in-situ data availability and higher absolute GPP than in mid-latitudes.

The comparison with in-situ GPP shows clear differences in performance of the VODCA2GPP model across different biomes. High performance is achieved in densely vegetated biomes while performance decreases in arid and less vegetated regions. A reason for the weaker performance in areas with less water availability might be adapted water regulation strategies of plants. Plants in drought-prone regions often reduce transpiration by limiting stomatal conductance to maintain a constant water potential even in times of extreme water scarcity (Sade et al., 2012). Since VOD is largely driven by the vegetation's water content, this isohydric behaviour of vegetation could at least partly explain relatively high VOD and consequently also overestimated GPP in those regions (Teubner et al., 2021).

Also, the observation bias which is introduced by unevenly distributed FLUXNET sites decreases the model's robustness. GPP is measured in situ only at a few locations and these stations are mostly located in temperate regions (e.g., Europe and North America) while semi-arid and tropical forest regions are underrepresented in the training data.

A comparison of uncertainties between VODCA2GPP and optical remote sensing based GPP (Xie et al., 2021) shows that in both cases topographic complexity decreases the reliability. Furthermore, the reliability of GPP estimates based on optical remote sensing is highly dependent on weather and illumination conditions. Clouds often contaminate or prevent the observations which is presumably the main reason why the largest uncertainties for FLUXCOM and MODIS are found in the wet tropics where GPP is known to be underestimated (de Almeida et al., 2018; Jung et al., 2020). In contrast, VODCA2GPP shows good skill for densely vegetated areas, including broadleaf evergreen forests. On the other hand, the relatively high uncertainties of VODCA2GPP in water-limited regions have not been reported for FLUXCOM or MODIS GPP, indicating that these are VOD-specific and presumably caused by the abovementioned isohydric behavior of plants in arid regions.

## 5.2 Limitations in VODCA and their impact on VODCA2GPP

Certain limitations in the VODCA v1 product exist, as outlined in Moesinger et al. (2020), which are partly also evident in VODCA v2 (Zotta et al., in prep) and thus propagate to VODCA2GPP. A known issue of VODCA v2 is caused by an observation gap between October 2011 and July 2012 for AMSR-E and AMSR2 (Table 1), which prevents a direct bias removal between the sensors. However, scaling between the sensors is achieved by using TMI observations North/South of 35°N/35S for X and Ku-band. Beyond these latitudes for X- and Ku-band, and globally for C-band, AMSR-E data were matched directly to AMSR2 using the last two years of AMSR-E and first two years of AMSR2 as reference period, under the assumption that trends between 2010-2014 are negligible (Moesinger et al., 2020). The result is that AMSR2 observations exhibit a slight positive bias in parts of North America which is also evident in a spatial break in VODCA v1 X- and Ku-Band trends (Moesinger et al., 2020). Although the impact of this procedure on VODCA2GPP trends is small and spatially limited, users are advised to keep the potential bias in mind when analysing VODCA2GPP data after 2012 for latitudes North/South of 35°N/35°S. Other limitations in VODCA concern the mixing of observations that were retrieved at different geometries (e.g., incidence angles) or observation times (Moesinger et al., 2020) and the data loss in certain regions, mostly in the Himalayas, which is caused by failure of the CDF-matching method due to insufficient input data (Moesinger et al., 2020). These issues, however, only have a small or spatially very limited influence on the final VODCA2GPP product. Furthermore, VOD retrievals exhibit a tendency for saturation in regions with very dense vegetation making it less likely to distinguish variability. A slight tendency for saturation was also observed for VODCA2GPP but the landcover based analysis exhibited a very high agreement between VODCA2GPP and in-situ GPP indicating high reliability of VODCA2GPP over densely vegetated regions.

Another limitation of VOD products in general, and thus also of VODCA2GPP, is the limited spatial resolution (0.25°). The lower achieved spatial granularity from passive microwave remote sensing is presumably another reason for the slightly weaker performance of VODCA2GPP in comparison with optical remote sensing derived GPP since the VODCA grid cell might not always be well represented by the in-situ measurements. Furthermore, the lower spatial resolution of VODCA2GPP is disadvantageous for the analysis of local GPP as small-scale variations in GPP might be hidden in VODCA2GPP. However, due to its long-term availability and generally high reliability, VODCA2GPP can still serve as a valuable source of data for various other applications (Sect. 5.4), especially concerning long-term climate-related studies and climate model evaluation.

## 5.3 Observed bias between VODCA2GPP and other remote sensing-based GPP datasets

There is only a very small bias when comparing VODCA2GPP with eddy-covariance measurements from FLUXNET but relatively large discrepancies in absolute GPP exist between VODCA2GPP and other remote-sensing-based products, and to a lesser extent with process-based TRENDY-v7 models. In tropical regions, the positive bias between VODCA2GPP and MODIS/FLUXCOM GPP can be partly explained by a reported and observed tendency of FLUXCOM and MODIS to

underestimate GPP in these regions (Turner et al. 2006; Wang et al., 2017; Fig. 1; Fig. A3). Outside the tropics, discrepancies in absolute GPP among products might be caused by the assumed overestimation of VODCA2GPP in winter months (i.e., in times with very little or no primary productivity). This overestimation is explained by water content in vegetation that is also present in these dormant periods. The sensitivity of microwaves to this water content results in non-zero VOD and, consequently, non-zero GPP (Teubner et al., 2021). This effect is similar to the isohydricity effect discussed in Sect. 5.1, which is an explanation for overestimation of VODCA2GPP in arid regions.

Another potential explanation for the positive bias of VODCA2GPP compared to MODIS/FLUXCOM is the presence of surface water and its impact on VOD retrievals. The presence of surface water is known to decrease the brightness temperature of the earth's surface and thus significantly decreases VOD retrievals (Bousquet et al., 2020). The impact of surface water contamination is evident in VODCA pixels that partly contain water bodies (e.g., lakes, rivers). These pixels exhibit systematically lower values than neighbouring pixels without water bodies. On the one hand, this leads to underestimation in the VODCA2GPP model in pixels containing surface water. On the other hand, it also has an effect on model training. This effect is caused by FLUXNET stations located close to water bodies, which hardly impact in-situ GPP retrievals but do cause erroneous VOD-retrievals at the 0.25° pixel scale. As a result, underestimated VOD is trained against unaffected in-situ GPP, which causes a slight but systematic global overestimation. A potential solution would be the masking of water-contaminated VOD. However, due to the constraints with temperature in the interaction term (eq. 3.3) this would strongly reduce the data available for training, which would potentially decrease the robustness of the VODCA2GPP model if the number of stations is not increased.

A general issue in the upscaling of GPP is the low availability of in-situ GPP, which is not only problematic in model training but also hampers a fair evaluation and validation at global scale. The remote sensing-based reference products, FLUXCOM and MODIS, however, are also trained and calibrated using in-situ GPP observations (Jung et al., 2020; Running et al., 1999) and can therefore not be viewed as fully independent from VODCA2GPP (Teubner et al., 2021). In contrast to observation-based GPP products, estimates from the TRENDY ensemble can be considered largely independent from VODCA2GPP.

## 5.4 Potential applications of VODCA2GPP

The validation results show that VODCA2GPP generally exhibits a high consistency with in-situ GPP observations and global state-of-the-art GPP products indicating that VODCA2GPP can be used complementary to current global GPP products. For the analysis of global as well as regional GPP anomalies, VODCA2GPP can provide valuable insights that might be hidden in other observational products due to the fundamentally differing observation methods and the associated limitations related to saturation effects, cloud cover, and other atmospheric effects such as water vapour content or aerosols (Xiao et al., 2019).

Also, for the monitoring of global GPP trends, VODCA2GPP has the potential to serve as independent and reliable source of data. The long-term trend analysis suggests that the majority of biomes have increased their primary productivity since 1988. There are several potential drivers for long-term increases in GPP, the most important ones being global warming, land-use changes and elevated $CO_2$ concentration in the atmosphere (Piao et al., 2019). The observed long-term trends in GPP across the different products support the theory of elevated atmospheric $CO_2$ leading to an increased uptake of $CO_2$ (Haverd et. al, 2020; Walker et al., 2020; Campbell et al., 2017; Schimel et al., 2015). The absence of trends in FLUXCOM does not contradict but rather supports this theory as FLUXCOM does not account for $CO_2$ fertilization effects (Jung et al., 2020). Due to the shortness of most in-situ GPP time series, it is, however, difficult to draw final conclusions on the existence, magnitude and reasons for long-term variations of GPP. Therefore, the global influence of atmospheric $CO_2$ on vegetation productivity remains uncertain, but VODCA2GPP allows to gain new perspectives on long-term GPP trends and might help to identify and quantify driving factors for increasing long-term primary productivity.

Furthermore, VODCA2GPP can be used as a largely independent source of data for the intercomparison and validation of other existing or newly developed global GPP datasets and models. Currently, a multitude of global GPP products exists showing large inconsistencies among products (Zhang and Aizhong, 2021). Similar to other global GPP datasets, VODCA2GPP cannot be seen as a true reference but using it as additional reference might help to acquire a more comprehensive picture on the performance of other datasets especially in the context of long- and short-term variability in GPP.

**6 Data Availability**

The VODCA2GPP data can be accessed (CC BY-NC-SA 4.0) at TU Wien Research Data under https://doi.org/10.48436/1k7aj-bdz35 (Wild et al., 2021).

**7 Conclusion**

In this dataset paper we introduced VODCA2GPP, a long-term GPP data record which uses multi-sensor, multi-frequency microwave VODCA data and temperature data from ERA5-Land for the upscaling of in-situ GPP from FLUXNET2015. The comparison of VODCA2GPP with FLUXNET in-situ GPP and global state-of-the-art GPP datasets showed good correspondence between the products in both the spatial and temporal domain, but with varying performance differences across biomes and analysed timescales. In tropical and arid regions, VODCA2GPP has significantly higher values than the reference datasets. Arid and mountainous areas were found to have the largest uncertainties. The analysis of monthly anomalies exhibited various extreme events in VODCA2GPP that are also found in one or more existing product indicating high plausibility of VODCA2GPP derived anomalies. Furthermore, trends derived from VODCA2GPP contain several plausible patterns that

match those derived from the TRENDY-v7 simulations but are not visible in both observational products and vice versa. This suggests that the novel microwave-based approach in VODCA2GPP has the potential to reveal novel findings about temporal dynamics in GPP at large scales that are not yet captured by other GPP products.

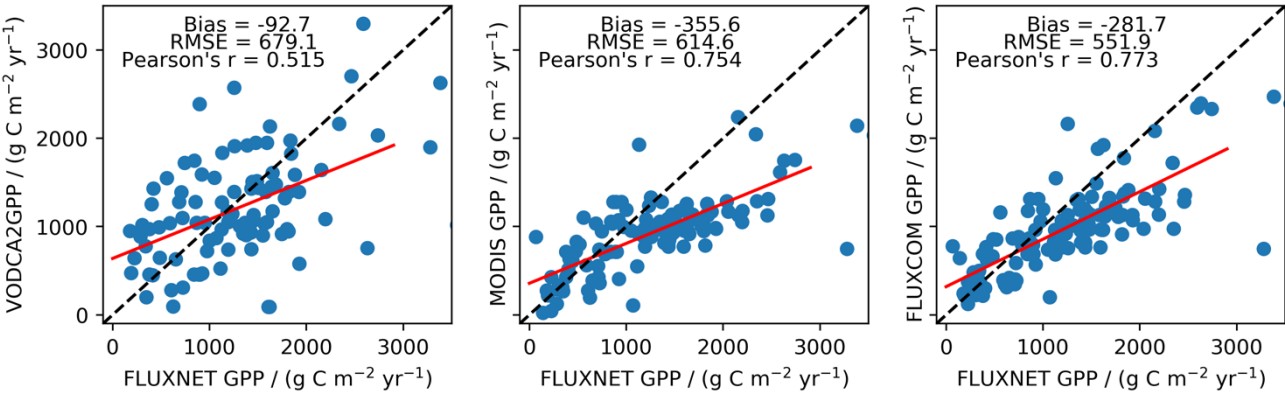

**Figure A1: Mean annual in-situ GPP (FLUXNET) plotted against mean annual GPP from VODCA2GPP, FLUXCOM and MODIS for the respective grid cells. Mean annual GPP was computed from all available overlapping years and thus each station is represented by one dot. Red lines indicate the best linear fits determined by ordinary linear regression and the black lines represent**
**the 1:1 lines.**

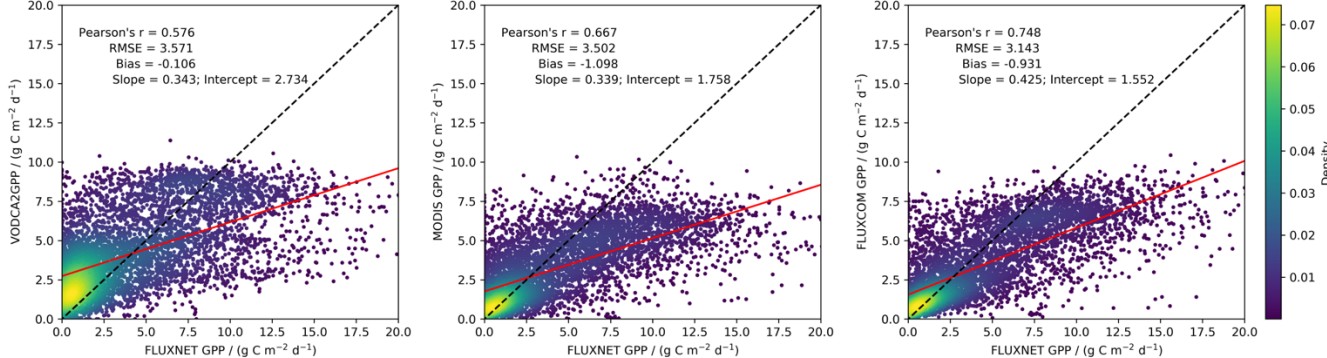

**Figure A2: GPP from FLUXNET plotted against GPP from VODCA2GPP, MODIS and FLUXCOM for the period 2002-2016 with monthly sampling.**

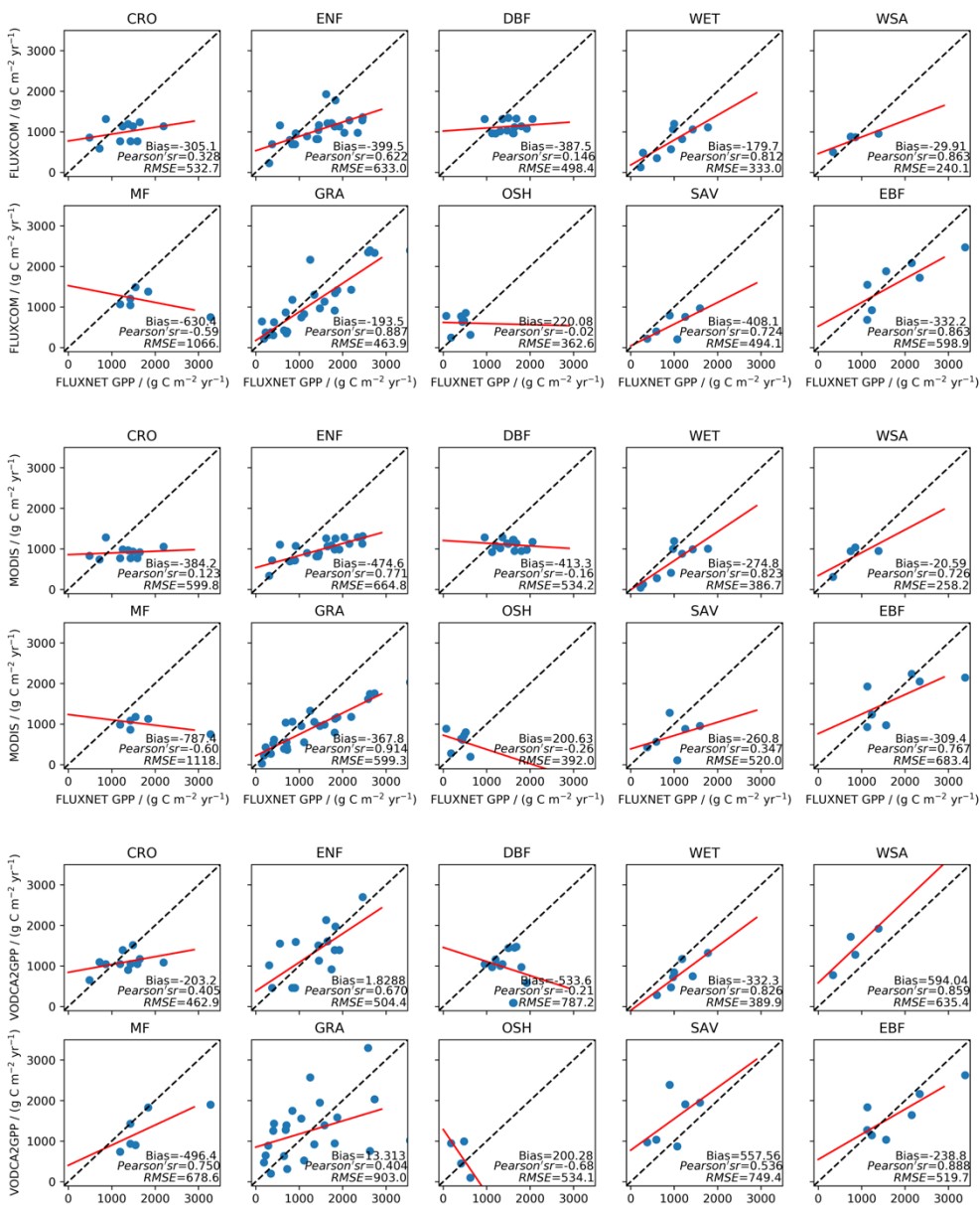

**Figure A3: Scatterplots of mean annual GPP for FLUXCOM, MODIS and VODCA2GPP for the period 2002-2016 per vegetation type. Vegetation types indicate the pre-dominant IGBP-vegetation type at the respective FLUXNET station.**
**Abbreviations: CRO: Croplands; ENF: Evergreen Needleleaf Forests; DBF: Deciduous Broadleaf Forests; WET: Permanent Wetlands; WSA: Woody Savannas; MF: Mixed Forests; GRA: Grasslands; OSH: Open Shrublands; SAV: Savannas; EBF:**
**Evergreen Broadleaf Forests**

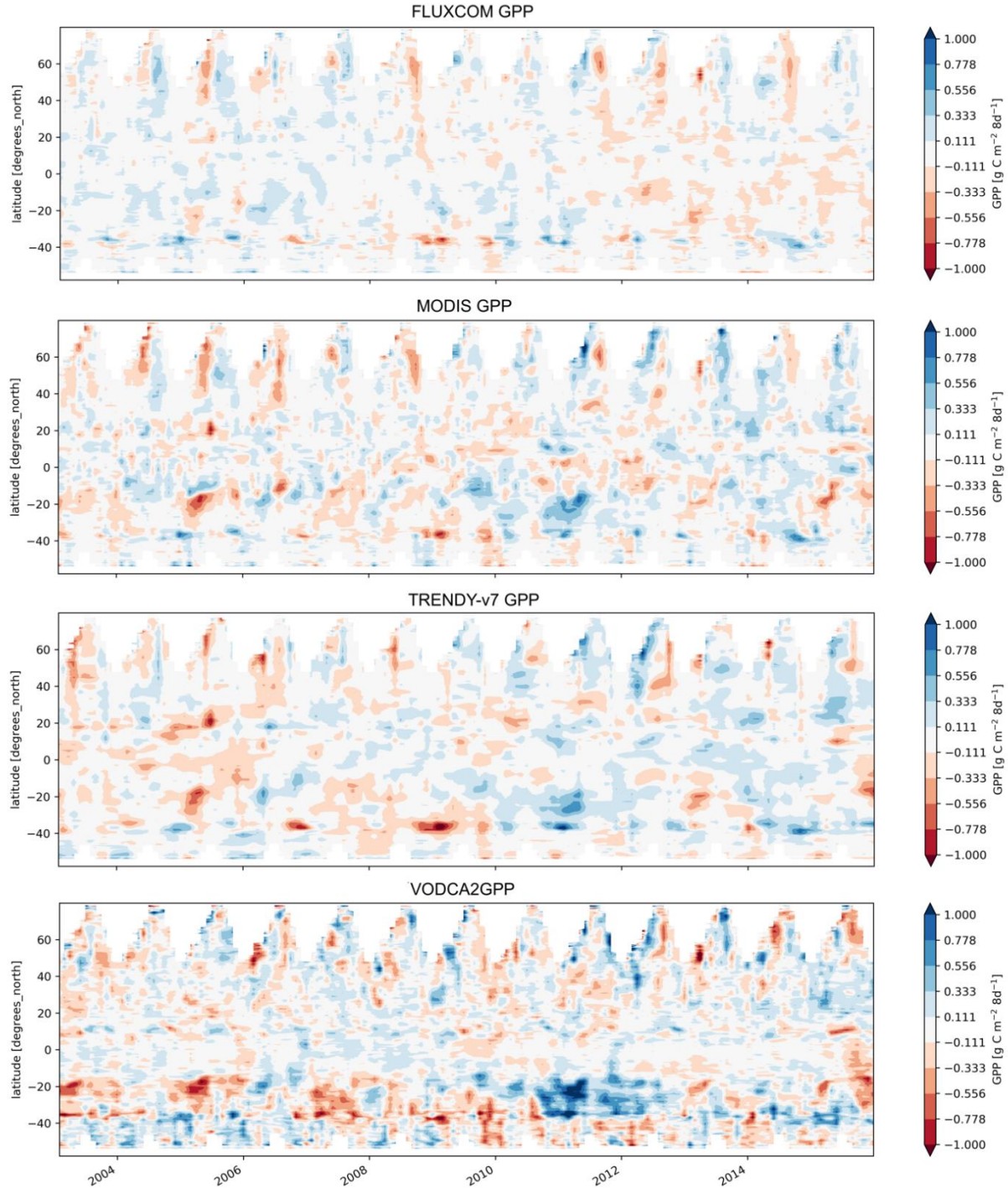

**Figure A4: Hovmoeller diagrams of monthly GPP-anomalies for each dataset.**

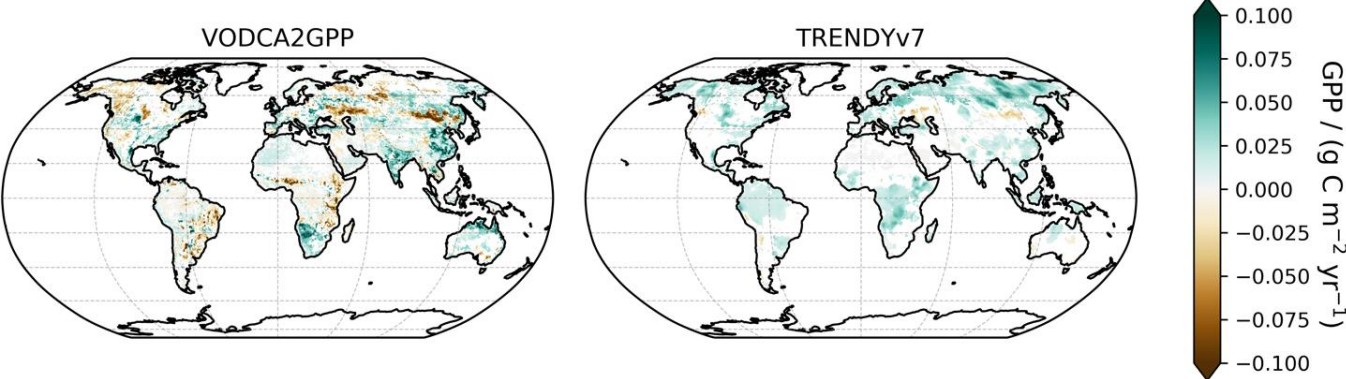

Figure A5: Global maps of yearly median GPP trends for the period 1988-2016 for VODCA2GPP and TRENDY-v7. White indicates non-significant trends.

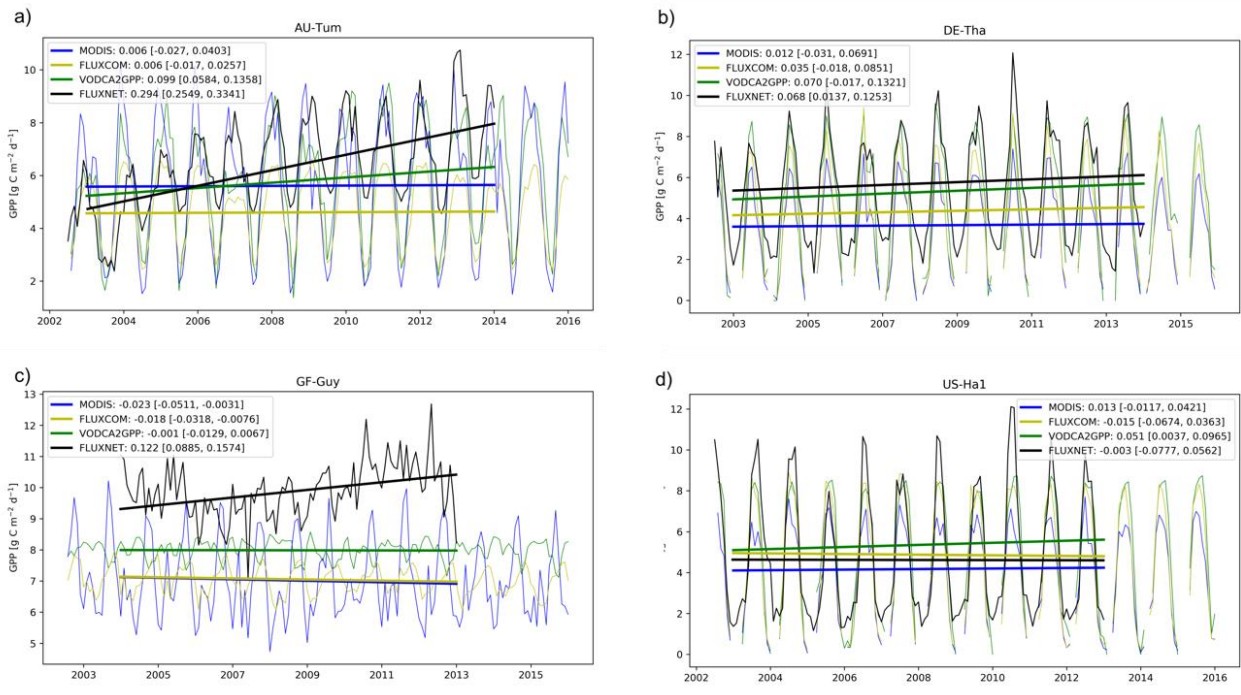

Figure A6: Exemplary collection of time series of in-situ FLUXNET GPP together with extracted time series from MODIS, FLUXCOM and VODCA2GPP. The stations were selected because of their high data availability for the respective landcover. The lines indicate the regression lines as obtained from the Theil-Sen slope estimation for yearly median GPP. The trends are computed for the common observation period with FLUXNET. The slope [g C m⁻² yr⁻¹] is depicted in the legend together with the respective 90% lower/upper confidence intervals. The depicted stations are:
a) AU-Tum: Tumbarumba, Australia; Lat: -35.65 °N, Lon: 148.15 °E; Landcover: EBF
b) DE-Tha: Tharandt, Germany; Lat: 50.96 °N, Lon: 13.57 °E; Landcover: ENF
c) GF-Guy: Guyaflux, French Guiana; Lat: 5.28 °N, Lon: -52.93 °E; Landcover: EBF
d)US-Ha1: Harvard Forest EMS Tower, United States; Lat: 42.54 °N, Lon: -72.17 °E; Landcover: DBF

**Table B1: Overview of FLUXNET Tier1 v1 stations within the period 1991-2014**

| FLUXNET ID | Name | Lon [°E] | Lat [°N] | Years used |
|---|---|---|---|---|
| AR-SLu | San Luis | -66.46 | -33.46 | 2009-2011 |
| AR-Vir | Virasoro | -56.19 | -28.24 | 2010-2012 |
| AT-Neu | Neustift | 11.32 | 47.12 | 2002-2012 |
| AU-ASM | Alice Springs | 133.25 | -22.28 | 2010-2013 |
| AU-Ade | Adelaide River | 131.12 | -13.08 | 2007-2009 |
| AU-Cpr | Calperum | 140.59 | -34.00 | 2010-2013 |
| AU-Cum | Cumberland Plains | 150.72 | -33.61 | 2012-2013 |
| AU-DaP | Daly River Savanna | 131.32 | -14.06 | 2008-2013 |
| AU-DaS | Daly River Cleared | 131.39 | -14.16 | 2008-2013 |
| AU-Dry | Dry River | 132.37 | -15.26 | 2008-2013 |
| AU-Emr | Emerald, Queensland, Australia | 148.47 | -23.86 | 2011-2013 |
| AU-Fog | Fogg Dam | 131.31 | -12.55 | 2006-2008 |
| AU-GWW | Great Western Woodlands, Wester Australia, Australia | 120.65 | -30.19 | 2013-2014 |
| AU-RDF | Red Dirt Melon Farm, Northern Territory | 132.48 | -14.56 | 2011-2013 |
| AU-Rig | Riggs Creek | 145.58 | -36.65 | 2011-2013 |
| AU-Rob | Robson Creek, Queensland, Australia | 145.63 | -17.12 | 2014 |
| AU-Tum | Tumbarumba | 148.15 | -35.66 | 2001-2013 |
| AU-Whr | Whroo | 145.03 | -36.67 | 2011-2013 |
| BE-Bra | Brasschaat | 4.52 | 51.31 | 2004-2013 |
| BE-Lon | Lonzee | 4.75 | 50.55 | 2004-2014 |
| BE-Vie | Vielsalm | 6.00 | 50.31 | 1996-2014 |
| BR-Sa3 | Santarem-Km83-Logged Forest | -54.97 | -3.02 | 2000-2004 |
| CA-NS1 | UCI-1850 burn site | -98.48 | 55.88 | 2001-2005 |
| CA-NS3 | UCI-1964 burn site | -98.38 | 55.91 | 2001-2005 |
| CA-NS4 | UCI-1964 burn site wet | -98.38 | 55.91 | 2002-2005 |
| CA-NS5 | UCI-1981 burn site | -98.49 | 55.86 | 2002-2005 |
| CA-NS6 | UCI-1989 burn site | -98.96 | 55.92 | 2001-2005 |
| CA-NS7 | UCI-1998 burn site | -99.95 | 56.64 | 2002-2005 |
| CA-Qfo | Quebec – Eastern Boreal, Mature Black Spruce | -74.34 | 49.69 | 2003-2010 |
| CA-SF1 | Saskatchewan – Western Boreal, forest burned in 1977 | -105.82 | 54.49 | 2003-2006 |
| CA-SF2 | Saskatchewan – Western Boreal, forest burned in 1989 | -105.88 | 54.25 | 2001-2005 |

| FLUXNET ID | Name | Lon [°E] | Lat [°N] | Year used |
|---|---|---|---|---|
| CA-SF3 | Saskatchewan – Western Boreal, forest burned in 1998 | -106.01 | 54.09 | 2001-2006 |
| CH-Cha | Chamau | 8.41 | 47.21 | 2006-2012 |
| CH-Fru | Frübüel | 8.54 | 47.12 | 2006-2012 |
| CH-Oe1 | Oensingen grassland | 7.73 | 47.29 | 2002-2008 |
| CN-Cha | Changbaishan | 128.10 | 42.40 | 2004-2005 |
| CN-Cng | Changling | 123.51 | 44.59 | 2007-2010 |
| CN-Dan | Dangxiong | 91.07 | 30.50 | 2004-2005 |
| CN-Din | Dinghushan | 112.54 | 23.17 | 2003-2005 |
| CN-Du2 | Duolun_grassland (D01) | 116.28 | 42.05 | 2006-2008 |
| CN-Ha2 | Haibei Shrubland | 101.33 | 37.61 | 2003-2005 |
| CN-HaM | Haibei Alpine Tibet site | 101.18 | 37.37 | 2002-2004 |
| CN-Qia | Qianyanzhou | 115.06 | 26.74 | 2003-2005 |
| CN-Sw2 | Siziwang Grazed (SZWG) | 111.90 | 41.79 | 2010-2012 |
| CZ-BK1 | Bily Kriz forest | 18.54 | 49.50 | 2004-2008 |
| CZ-BK2 | Bily Kriz grassland | 18.54 | 49.49 | 2004-2006 |
| DE-Akn | Anklam | 13.68 | 53.87 | 2009-2014 |
| DE-Gri | Grillenburg | 13.51 | 50.95 | 2004-2014 |
| DE-Hai | Hainich | 10.45 | 51.08 | 2000-2012 |
| DE-Kli | Klingenberg | 13.52 | 50.89 | 2004-2014 |
| DE-Lkb | Lackenberg | 13.30 | 49.10 | 2009-2013 |
| DE-Obe | Oberbärenburg | 13.72 | 50.78 | 2008-2014 |
| DE-RuS | Selhausen Juelich | 6.45 | 50.87 | 2011-2014 |
| DE-Spw | Spreewald | 14.03 | 51.89 | 2010-2014 |
| DE-Tha | Tharandt | 13.57 | 50-96 | 1996-2014 |
| DK-NuF | Nuuk Fen | -51.39 | 64.13 | 2008-2014 |
| DK-Sor | Soroe | 11.64 | 55.49 | 1996-2012 |
| ES-LgS | Laguna Seca | -2.97 | 37.10 | 2007-2009 |
| ES-Ln2 | Lanjaron-Salvage logging | -3.48 | 36.97 | 2009 |
| FI-Hyy | Hyytiala | 24.30 | 61.85 | 1996-2014 |
| FI-Jok | Jokioinen | 23.51 | 60.90 | 2000-2003 |
| FR-Gri | Grignon | 1.95 | 48.84 | 2004-2013 |

| FLUXNET ID | Name | Lon [°E] | Lat [°N] | Year used |
|---|---|---|---|---|
| FR-Pue | Puechabon | 3.60 | 43.74 | 2000-2013 |
| GF-Guy | Guyaflux (French Guiana) | -52.92 | 5.28 | 2004-2012 |
| IT-CA1 | Castel d'Asso 1 | 12.03 | 42.38 | 2011-2013 |
| IT-CA2 | Castel d'Asso 2 | 12.03 | 42.38 | 2011-2013 |
| IT-CA3 | Castel d'Asso 3 | 12.02 | 42.38 | 2011-2013 |
| IT-Cp2 | Castelporziano 2 | 12.36 | 41.70 | 2012-2013 |
| IT-Isp | Ispra ABC-IS | 8.63 | 45.81 | 2013-2014 |
| IT-Lav | Lavarone | 11.28 | 45.96 | 2003-2012 |
| IT-Noe | Arca di Noé – Le Prigionette | 8.15 | 40.61 | 2004-2012 |
| IT-PT1 | Parco Ticino forest | 9.06 | 45.20 | 2002-2004 |
| IT-Ren | Renon | 11.43 | 46.59 | 1998-2013 |
| IT-Ro1 | Roccarespampani 1 | 11.93 | 42.41 | 2000-2008 |
| IT-Ro2 | Roccarespampani 2 | 11.92 | 42.39 | 2003-2012 |
| IT-SR2 | San Rossore 2 | 10.29 | 43.73 | 2013-2014 |
| IT-SRo | San Rossore | 10.28 | 43.73 | 1999-2012 |
| IT-Tor | Torgnon | 7.58 | 45.84 | 2008-2013 |
| JP-MBF | Moshiri Birch Forest Site | 142.32 | 44.39 | 2003-2005 |
| JP-SMF | Seto Mixed Forest Site | 137.08 | 35.26 | 2002-2006 |
| NL-Hor | Horstermeer | 5.07 | 52.24 | 2004-2011 |
| NL-Loo | Loobos | 5.74 | 52.17 | 1996-2013 |
| NO-Adv | Adventdalen | 15.92 | 78.19 | 2012-2014 |
| RU-Che | Cherski | 161.34 | 68.61 | 2002-2005 |
| RU-Cok | Chokurdakh | 147.49 | 70.83 | 2003-2013 |
| RU-Fyo | Fyodorovskoye | 32.92 | 56.46 | 1998-2013 |
| RU-Ha1 | Hakasia steppe | 90.00 | 54.73 | 2002-2004 |
| SD-Dem | Demokeya | 30.48 | 13.28 | 2005-2009 |
| US-AR1 | ARM USDA UNL OSU Woodward Switchgrass 1 | -99.42 | 36.43 | 2009-2012 |
| US-AR2 | ARM USDA UNL OSU Woodward Switchgrass 2 | -99.60 | 36.64 | 2009-2012 |
| US-ARM | ARM Southern Great Plains site - Lamont | -97.49 | 36.61 | 2003-2012 |
| US-Blo | Blodgett Forest | -120.63 | 38.90 | 1997-2007 |
| US-Ha1 | Harvard Forest EMS Tower (HFR 1) | -72.17 | 42.54 | 1991-2012 |

| FLUXNET ID | Name | Lon [°E] | Lat [°N] | Year used |
|---|---|---|---|---|
| US-Los | Lost Creek | -89.98 | 46.08 | 2000-2014 |
| US-MMS | Morgan Monroe State Forest | -86.41 | 39.32 | 1999-2014 |
| US-Me6 | Metolius Young Pine Burn | -121.61 | 44.32 | 2010-2012 |
| US-Myb | Mayberry Wetland | -121.77 | 38.05 | 2011-2014 |
| US-Ne1 | Mead – irrigated continuous maize site | -96.48 | 41.17 | 2001-2013 |
| US-Ne2 | Mead – irrigated maize-soybean rotation site | -96.47 | 41.16 | 2001-2013 |
| US-Ne3 | Mead – rainfed maize-soybean roatation site | -96.44 | 41.18 | 2001-2013 |
| US-SRM | Santa Rita Mesquite | -110.87 | 31.82 | 2004-2014 |
| US-Syv | Sylvania Wilderness Area | -89.35 | 46.24 | 2001-2014 |
| US-Ton | Tonzi Ranch | -120.97 | 38.43 | 2001-2014 |
| US-Tw3 | Twitchell Alfalfa | -121.65 | 38.12 | 2013-2014 |
| US-UMd | UMBS Disturbance | -84.70 | 45.56 | 2007-2014 |
| US-Var | Vaira Ranch-Ione | -120.95 | 38.41 | 2000-2014 |
| US-WCr | Willow Creek | -90.08 | 45.81 | 1999-2014 |
| US-Whs | Walnut Gulch Lucky Hills Shrub | -110.05 | 31.74 | 2007-2014 |
| US-Wkg | Walnut Gulch Kendall Grasslands | -109.94 | 31.74 | 2004-2014 |
| ZM-Mon | Mongu | 23.25 | -15.44 | 2007-2009 |

## Author Contributions

BW curated the dataset, carried out the analysis and drafted the manuscript with inputs from all authors. IT, MF and WD created and designed the model. IT provided the model implementation and contributed to data preparation. LM, RZ and RS produced the VODCA data. SS provided the TRENDY-v7 ensemble data. All authors contributed to discussions about the
565 methods and results and provided feedback and input for the paper.

## Competing interests

The authors declare that they have no conflict of interest.

**Acknowledgements**

Benjamin Wild, Ruxandra-Maria Zotta and Leander Moesinger are supported through the EOWAVE project funded by the

570 TU Wien Wissenschaftspreis 2015, which was awarded to Wouter Dorigo https://climers.geo.tuwien.ac.at/climers/research/vegetation/eowave/. This work used eddy covariance data acquired and shared by the FLUXNET community, including these networks: AmeriFlux, AfriFlux, AsiaFlux, CarboAfrica, CarboEuropeIP, CarboItaly, CarboMont, ChinaFlux, Fluxnet-Canada, GreenGrass, ICOS, KoFlux, LBA, NECC, OzFlux-TERN, TCOS-Siberia, and USCCC. The FLUXNET eddy covariance data processing and harmonization was carried out by

575 the ICOS Ecosystem Thematic Center, AmeriFlux Management Project and Fluxdata project of FLUXNET, with the support of CDIAC, and the OzFlux, ChinaFlux and AsiaFlux offices.

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

list): VODCA v2: An improved multi-sensor and frequency vegetation optical depth dataset for long-term vegetation monitoring (preliminary title), in preparation.