# Peer review of "VODCA2GPP - A new global, long-term (1988-2020) GPP dataset from microwave remote sensing"

_Earth System Science Data, 2021_

## Author Comment (AC2)

Reviewer comments

Reply to comments

General comments:

This study introduced a new global GPP dataset using satellite-derived vegetation optical depth (VOD) dataset spanning 1988-2020 in spatial resolution of 0.25o. The method for relating GPP to VOD was proposed in Teubner et al (2019, 2021). This study also presented validation against FLUXNET observation and other global GPP datasets (MODIS, FLUXCOM, TRENDY models). The difference of spatial and temporal GPP dynamics from other GPP reference datasets were shown. However, given the lower VOD-GPP model performance and extremely high global amount of VODCA2GPP, I do have concern about the quality of this dataset and its advantage over other datasets. The writing also needs a lot of improvements since there are both phrasing mistakes and technical mistakes.

Response: Dear Referee,

Thank you very much for reviewing our paper and giving very constructive feedback. We understand the concerns regarding the quality of the VODCA2GPP dataset but we strongly believe that although VODCA2GPP shows a slightly weaker performance than current state-of-the-art GPP products when comparing mean annual GPP, VODCA2GPP can still be a very valuable data source for studying the global productivity.

It is true that VODCA2GPP GPP estimates are relatively high compared to estimates from current state-of-the-art products. On the other hand, MODIS and FLUXCOM are known to substantially underestimate GPP especially in very productive regions (Fig 3 (new), Fig A1 (new); Turner et al. 2006; Wang et al., 2017) which is why large differences between MODIS/FLUXCOM and VODCA2GPP do not necessarily implicate substantial overestimation of VODCA2GPP. In fact, Welp at al. (2011) suggested that current estimates of GPP are generally too low and they estimated the actual yearly GPP to be between 150-175 Pg C yr$^{-1}$. Koffi et al. (2012) came to similar conclusions (146 ± 19 Pg C yr$^{-1}$). The comparison of mean annual GPP between VODCA2GPP and FLUXNET GPP does not suggest an overall overestimation of VODCA2GPP but exhibits lowest bias for VODCA2GPP among all analyzed datasets (Fig 3 (new)). Nevertheless, we agree that VODCA2GPP has a tendency for overestimation for arid regions (e.g., (woody) savannas and open shrubland; Fig A1 (new)). However, due to the missing consensus among existing datasets (Anav et al., 2015) we do not see this as an inherent flaw of VODCA2GPP but rather as a systematic problem in the observation of GPP originating from generally low in-situ data availability.

Regarding the style of writing we are very grateful for your detailed feedback and we will thoroughly revise the manuscript and improve the writing and other technical mistakes that are pointed out in your review.

Specific comments:

The global total amount of VODCA2GPP datasets is not given.

The global total GPP as derived from VODCA2GPP amounts on average to 200 ± 2.2 Pg C yr$^{-1}$. We will add this number in a revised version of the manuscript.

line 19, line31, it should be 'at global scale'.

Thank you for this suggestion, we have revised this in the latest version.

line 32, it should be 'different'.

Thank you for this suggestion, we have revised this in the latest version.

line 32-33, it should be 'similar significant increase'. Please check the usage of two 'for'.

Thank you for this suggestion, we have revised this in the latest version.

line 43-44, please reformulate it.

Thank you for this suggestion, we have revised this in the latest version.

line 46-47, please reformulate it.

Thank you for this suggestion, we have revised this in the latest version.

line 51, please use clause.

Thank you for this suggestion, we have revised this in the latest version.

line 53, what did you mean by 'measurement bias'? Is it affected by the distribution of flux towers or just measurement techniques?

Here, we refer to the uneven distribution of flux towers which leads to the under/overrepresentation of certain landcovers/biomes in the training data. We have revised this in the latest version.

line 236, please reformulate it.

Thank you for this suggestion, we have revised this in the latest version.

line 245, for site-level validation, you should also use cross validation method.

We will include a 10-fold cross-validation analysis in addition to the uncertainty analysis.

line 264, the subtitle for section 4 is missing.

Thank you for making us aware of the missing subtitle. We have added the subtitle in the latest version.

In figure 2, the uncertainty is very high in desert area (more than ten time of the GPP itself, why?), for example, northern Africa, which is blank in figure 1. Please use single-hue color series and check to ensure the consistency with figure 1.

It is true that the uncertainties are very high in some very arid regions such as the Sahara. The exact reason for the low robustness of the model in these regions is unclear but we assume that these instabilities are partly explicable with missing training data from desert areas.

The reason for the missing consistency between Figure 1a and Figure 2 (old) is that in Figure 1a we only used data that is available in all three products (MODIS, FLUXCOM, VODCA2GPP) in order to allow comparability. We agree that this is confusing and added an additional figure including all available VODCA2GPP data which is consistent with Figure 2 (old).

Furthermore, we updated Figure 2 by using a single-hue color scale and merged it with the added figure of VODCA2GPP mean annual GPP.

[Figure]

Figure 2 (new): a) Mean annual GPP as derived from VODCA2GPP for the period 1988-2019. b) Standard deviation of mean yearly annual GPP (1988-2019) as obtained by the uncertainty analysis.

In figure 3, please check the unit in both x and y axis. Please use dashed line for 1:1 line. Here, you presented the site-level evaluation for mean annual GPP. What about the annual, seasonal (or monthly) evaluation? In fact, Pearson value of around 0.5 seems to be relatively lower for GPP estimation. Did you try 10-fold cross validation for model performance?

In Figure 3, we adapted the units, replaced the 1:1 line and added the bias in the performance metrics:

[Figure]

Figure 3 (new): Mean annual in-situ GPP (FLUXNET) plotted against mean annual GPP from VODCA2GPP, FLUXCOM and MODIS for the respective grid cells. Mean annual GPP was computed from all available overlapping years and thus each station is represented by one dot. Red lines indicate the best linear fits determined by ordinary linear regression and the black lines represent the 1:1 lines.

We agree that it would also make sense to use annual, seasonal or even monthly values for the comparison. A slight advantage of using mean annual values is that it can be visualized without a density map while still including the entire information from the data.

VODCA2GPP underperforms mainly for sparsely vegetated biomes, while performance is good for densely vegetated regions (Fig A1). In these regions VODCA2GPP's performance metrics (Pearson's r and RMSE) are comparable or even better than those from MODIS and FLUXCOM. Therefore, we suggest that VODCA2GPP can be used as a reliable source of GPP data as long as the users are aware of the performance differences depending on the landcover. We understand that this can be seen as drawback but considering the fact that VODCA2GPP performs well for most regions and is largely independent from current state-of-the-art products, we believe that VODCA2GPP can provide very valuable input for the scientific community. Due to the independence of VODCA2GPP, it also allows users to gain insights on global GPP trends and anomalies that are partly hidden in optical remote sensing based GPP datasets.

We will include 10-fold cross validation in a revised version of the manuscript.

line 308, when you mention 'good temporal agreement', you need to specify the region, since the correlation is even negative in tropical America.

We agree that this needs to be reformulated more precisely. We will revise this.

line 308-309, it should be 'the highest' and 'the lowest'.

Thank you for this suggestion, we have revised this in the latest version.

line 311, please rephrase it.

Thank you for this suggestion, we have revised this in the latest version.

line 313, why monthly GPP anomalies are compared? Why you did not use monthly GPP instead?

Here, we wanted to evaluate VODCA2GPP's ability to capture extreme events and anomalies in general.

line 334, here you used 2002-2016. Why did you use 2003-2015 in next paragraph?

Thank you for bringing this to our attention. The period 2003-2015 is correct. We will revise this.

line 335, there are only three sentences in this paragraph. It would be better to merge it with next paragraph.

We will merge the paragraphs.

In figure 6, at global scale, the VODCA2GPP is two times of the MODIS and FLUXCOM. So the global GPP can be more than 200 PgC yr-1, which is the highest estimation as far as I know. You should give reasonable evidence for such value, or it won't be convincing.

It is true that VODCA2GPP estimates are substantially higher than MODIS and FLUXCOM estimates but as already outlined above, optical products are known to significantly underestimate GPP, especially in very productive regions (Fig 3 (new), Fig A1 (new); Turner et al., 2006; Wang et al., 2017). Thus, differences in global annual GPP between VODCA2GPP and MODIS/FLUXCOM need to be interpreted with caution. In fact, the comparison of VODCA2GPP with in-situ GPP does not suggest a systematic overestimation (Fig 3 (new)). VODCA2GPP only shows strong overestimation for (woody) savannas and open shrublands (Fig A1 (new)). For all FLUXNET sites located in other landcovers our analysis suggests very mild overestimation or even underestimation of VODCA2GPP. Furthermore, the comparison of VODCA2GPP with the mean of all TRENDY models (Fig. 6) indicates that the overestimation is by far not as severe as implied by MODIS and FLUXCOM.

In figure 7, it should be 2002-2016 or 2003-2015?

2003-2015 is correct.

In section 5.2, TRENDY ensemble is indeed independent from VODCA2GPP. But I couldn't get the meaning of this paragraph. The process-based estimation without accounting for nutrient constraints is also reported to overestimate GPP.

The objective of this section is to emphasize that the validation of global GPP datasets generally needs to be interpreted with caution due to a) the generally low availability and uneven distribution of FLUXNET stations and b) due to the often shared in-situ data-source (i.e. FLUXNET). MODIS, FLUXCOM and VODCA2GPP use the same reference measurements for calibration and validation. As a result MODIS and FLUXCOM cannot be viewed as completely independent from VODCA2GPP although they are using fundamentally differing modelling approaches. Thus, this paragraph specifically aims to transparently inform potential users about these inevitable validation problems. We will merge this paragraph with section 5.4.

In section 5.6, in terms of the VODCA2GPP datasets, more similarities with TRENDY than that of other observational products were mentioned. Is it really a strong point for this GPP dataset?

As already mentioned in response 13, one strong argument for VODCA2GPP is its large independence from other (optical) observational products. This independence is evident especially when analyzing long-term trends in GPP where process-based and optical remote sensing products only exhibit very weak correspondence. VODCA2GPP on the other hand exhibits patterns that are visible in at least one other dataset which led us to the conclusion that VODCA2GPP is able to largely bridge the obvious gap between process-based and optical remote-sensing based products. Thus, not the good correspondence with TRENDY alone is a strong point for VODCA2GPP but more importantly its ability to complement existing products, both process-based and observational, is advantageous.

In figure A1, please check the unit in both x and y axis. You can give annual, seasonal (or monthly) evaluation as well.

We adapted the units in the axis labelling and added the bias in the performance metrics:

[Figure]

Figure A1 (new): Scatterplots of mean annual GPP for the period 2002-2016 per vegetation type. Vegetation types indicate the predominant IGBP-vegetation type at the respective FLUXNET station. Abbreviations: CRO: Croplands; ENF: Evergreen Needleleaf Forests; DBF: Deciduous Broadleaf Forests; WET: Permanent Wetlands; WSA: Woody Savannas; MF: Mixed Forests; GRA: Grasslands; OSH: Open Shrublands; SAV: Savannas; EBF: Evergreen Broadleaf Forests.

The code for producing the data and figures should also be publicly available in a repository for transparency reason.

We generally embrace the idea of open access science which is why we made the VODCA2GPP dataset publicly and freely available. However, at the moment we do not consider our work finished. We still strive for improvements of the VODA2GPP dataset which we will then of course also share with the scientific (and non-scientific) community. Therefore, we are currently not planning to make the VODCA2GPP code publicly available but we are happy to share the code with anyone who approaches us with ideas for collaboration. Regarding the transparency, we described the model algorithms and the data that is depicted in the figures in detail which is why we do not see a necessity to explicitly share the underlying code.

References
Anav, A., et al. (2015), Spatiotemporal patterns of terrestrial gross primary production: A review, Rev. Geophys., 53, 785– 818, doi:10.1002/2015RG000483.

Koffi, E. N., Rayner, P. J., Scholze, M., and Beer, C. (2012), Atmospheric constraints on gross primary productivity and net ecosystem productivity: Results from a carbon-cycle data assimilation system, Global Biogeochem. Cycles, 26, GB1024, doi:10.1029/2010GB003900.

Turner, D. P., Ritts, W. D., Cohen, W. B., Gower, S. T., Running,S. W., Zhao M., Costa, M. H., Kirschbaum, A. A., Ham, J. M.,Saleska, S. R., and Ahl, D. E.: Evaluation of MODIS NPP and GPP products across multiple biomes, Remote Sens. Environ.,102, 282–292, https://doi.org/10.1016/j.rse.2006.02.017, 2006.

Wang, L., Zhu, H., Lin, A., Zou, L., Qin, W., and Du, Q.: Evaluation of the Latest MODIS GPP Products across Multiple 770Biomes Using Global Eddy Covariance Flux Data, Remote Sensing, 9, 5, https://doi.org/10.3390/rs9050418, 2017.

Welp, L. R., Keeling, R. F., Meijer, H. A., Bollenbacher, A. F., Piper, S. C., Yoshimura,K., Francey, R. J., Allison, C. E., and Wahlen, M.:Interannual variability in the oxygen isotopes of atmospheric CO2 driven by El Niño. Nature, 477(7366):579–582, https://doi.org/10.1038/nature10421, 2011.

---

## Author Comment (AC3)

**Important additional remark to AC1 and AC2:**

We realized just after submission of our author comments that Figure 2b (standard deviation) is not scaled correctly. We apologize for this mistake and we updated the manuscript accordingly. The updated Figure 2 can be found below:

[Figure]

Figure 2 (new): a) Mean annual GPP as derived from VODCA2GPP for the period 1988-2019. b) Standard deviation of mean yearly annual GPP (1988-2019) as obtained by the uncertainty analysis.

---

## Author Response (AR1)

Formatting as follows:

Reviewer comments
Reply to comments
Changes to the manuscript

**Anonymous Referee**

This study produced a global GPP product (VODCA2GPP) using satellite-based vegetation optical depth (VOD) datasets from 1988-2020. Basically, the authors used the method proposed by Teubner et al. (2019), and evaluated the product accuracy based on the measurements of multiple eddy covariance towers globally. In addition, the authors compared the spatial and temporal differences of VODCA2GPP with MODIS and FLUXCOM GPP products. There are several main concerns in my mind to make me suggest rejecting the manuscript because of the important and inherent flaws of VOD data.

*Please note: All references to materials (e.g. Figures, Tables, Sections) now refer to the numbering of the revised manuscript.*

Response (1): Dear Referee,

Many thanks for reviewing our paper and outlining various concerns. We believe that most of the mentioned issues can be clarified which we will try to do in this reply.

To begin with, we would like to emphasize that the method that was used for producing the VODCA2GPP dataset is not solely based on Teubner et al. (2019) as suggested by your review but, even more importantly, on the refined methods from Teubner et al. (2021). The main difference between these two modelling approaches is that Teubner et al. (2021) additionally include air temperature in their model formulation which improves model performance substantially, especially in terms of temporal variability which is explained in detail in the manuscript.

First, VOD indicates the vegetation biomass, a large carbon pool, but the GPP indicates one of the carbon fluxes, much smaller compared to biomass. Teubner et al. (2019) and this study used the changes of VOD to simulate GPP, which requires the high performance of VOD to indicate biomass changes, a huge challenge. GPP is sensitive to leaf area index or leaf biomass, not wood and root biomass. However, leaf biomass is quite small fraction compared to other two components. It is obviously using VOD to estimate GPP has much large uncertainty.

Response (2): VOD is generally a strong indicator for vegetation water content which also makes it closely related to vegetation biomass. However, the sensitivity of VOD to different biomass compartments depends also on the frequency of the microwave band that is use to retrieve VOD. While longer wavelengths (e.g. L-Band) are mainly sensitive to large plant structures (i.e. tree trunks and large branches), shorter wavelengths (C/X/Ku-Band) provide information on smaller vegetation structures such as leaves and twigs (Frappart et al., 2020; Woodhouse, 2005). Hence high-frequency VOD is also strongly related to leaf area index and NDVI (Li et al., 2021), which are widely used predictors (among others) for vegetation productivity. Thus, in theory, high-frequency VOD and changes in VOD (ΔVOD) are good indicators for GPP. Teubner et al. (2018) demonstrated in their data-driven study that GPP is indeed highly correlated with high-frequency VOD and ΔVOD from X-

and C-band. Based on these findings they developed the VOD2GPP model (Teubner et al. 2019, 2021) which incorporates VOD, ΔVOD, air temperature and the temporal median VOD per grid cell. The latter helps to deduct a static vegetation component which makes the model even more sensitive to small-scale changes in biomass (more on the mdn(VOD)-term (including changes to the manuscript) in Response 5).

Especially based on the relationship between high-frequency VOD and canopy biomass, it is not "obvious" that VOD-based GPP estimates have much higher uncertainties but it is a new and promising method. Especially, the use of VOD might overcome limitations of vegetation variables from optical sensors (e.g. cloud cover) in observing GPP and provide the scientific community with GPP estimates that allow to gain new insights on the global carbon cycle as VODCA2GPP is largely independent from existing products.

We agree with the reviewer that the used VOD datasets require a high performance. Therefore, we used the novel VOD Climate Archive (VODCA) dataset which has been used in several studies and is highly appreciated in the scientific community (e.g. Li et al., 2021; Frappart et al., 2020). For the production of VODCA2GPP, we specifically used VODCA from the high-frequency C-, X- and Ku-bands in order to account primarily for changes in the canopy biomass Additionally, we are using very short time intervals (mostly only a few days) for the computation of ΔVOD. With such small time steps, we are further reducing the influence of woody components as they are not expected to show as strong changes as leaf biomass.

Page 7, line 201: "*The usage of short time intervals (in the order of several days) for the computation of Δ(VOD) is crucial since it reduces the influence of larger vegetation components (e.g., stems) and makes the model more sensitive to changes in leaf biomass.*"

From the validation results, we found low model performance of VODCA2GPP shown as fig. 3a. VODCA2GPP only reproduce spatial variations of GPP about 25% (pearson r = 0.515), and I can't trust the information provided by VODCA2GPP as such low performance.

Response (3): We agree that the performance of VODCA2GPP in comparison to the mean annual GPP at site level is lower compared to the established optical products MODIS and FLUXCOM. However, this performance of VODCA2GPP strongly depends on land cover (Fig A3). VODCA2GPP shows a good agreement with mean annual GPP at sites in wetlands, woody savannas, mixed forests, and (remarkably) evergreen broadleaved forests. The performance is rather low at sites with deciduous broadleaved forests, grasslands and open shrublands (Fig A3). Furthermore, differences in performance regarding absolute deviations (expressed in RMSE) are small. Therefore, we do not agree that VODCA2GPP cannot be generally trusted, but on the contrary we believe that based on our results VODCA2GPP delivers very reliable and largely independent information on global productivity. Additionally, we again want to highlight the overall good correspondence with the temporal dynamic of GPP from MODIS and FLUXCOM (Fig. 3) which shows that VODCA2GPP is trustworthy and of similar quality as the optical products.

We added a cross-validation and a more thorough comparison with in-situ GPP for different time-scales to give a better overview on the quality of VODCA2GPP (please see response 9)

In addition, limited by VOD products, the spatial resolution of VODCA2GPP is only 0.25°, which is not far enough for scientific applications, as MODIS Landsat GPP has reached to 30 meter spatial resolution.

Response (4): The applicability of the VODCA2GPP product depends on the type of application. Analogously like in remote sensing of soil moisture, we all know that soil moisture shows a very high

spatial variability depending on topography and soil conditions but still satellite-derived soil moisture datasets at 0.25° spatial resolution receive a wide scientific application and are especially suitable for long-term climate-oriented studies (e.g. Dorigo et al. 2017). Despite the availability of some GPP products at higher spatial resolution, to the best of our knowledge there does not exist a global GPP dataset with a spatial resolution of 30 meters. The GPP product with the finest resolution that is available globally is MODIS GPP with 500m. FLUXCOM reaches 10km and the TRENDY models are available at 1°. Hence the spatial resolution of the VODCA2GPP product, might be well suited for comparisons with simulations from global vegetation and carbon cycle models.

We added a paragraph in Sect. 5.2 where we discuss the limitations associated with the lower resolution and why VODCA2GPP can still be a valuable source of data:

Page 19, line 427: *"Another limitation of VOD products in general, and thus also of VODCA2GPP, is the limited spatial resolution (0.25°). The lower achieved spatial granularity from passive microwave remote sensing is presumably another reason for the slightly weaker performance of VODCA2GPP in comparison with optical remote sensing derived GPP since the VODCA grid cell might not always be well represented by the in-situ measurements. Furthermore, the lower spatial resolution of VODCA2GPP is disadvantageous for the analysis of local GPP as small-scale variations in GPP might be hidden in VODCA2GPP. However, due to its long-term availability and generally high reliability, VODCA2GPP can still serve as a valuable source of data for various other applications (Sect. 5.4), especially concerning long-term climate-related studies and climate model evaluation."*

Second, I am confused to model algorithms proposed by this study. The algorithms of this study look different a little bit with those of Teubner et al. (2019). Equ. 3.2 represents the model algorithms, the authors added the third term the temporal median of VOD (mdn(VOD)) derived from the complete time series which serves as a proxy for the landcover. Unfortunately, the authors did not explain how mdn(VOD) can indicate landcover. And I also concern why the authors need this term for simulating GPP.

Response (5): The model algorithms in this study are different from those in Teubner et al. (2019) because we make use of the refined model in Teubner et al. (2021). However, the mdn(VOD) term was already incorporated and described in the model algorithms from Teubner et al. (2019) representing a static vegetation biomass component for each pixel which we translated as 'landcover proxy'. We agree that this term is a slight simplification that can lead to misunderstandings.

The reason for incorporating mdn(VOD) in the model is to subtract larger structural components making the model more closely related to smaller structures like leaves, which increases model performance (Teubner et al., 2019). We will include a more detailed explanation on the mdn(VOD) term in a revised version of the manuscript.

We replaced the term 'landcover' with 'vegetation density' and added a more detailed explanation of the mdn(VOD) term:

*Page 7, line 182: (…) the temporal median of VOD (mdn(VOD)) derived from the complete time series which serves as a proxy of vegetation density. Specifically, mdn(VOD) is incorporated to subtract larger structural vegetation components which makes the resulting model more closely related to biomass changes of smaller structural vegetation components such as leaves. It was shown that this increases model performance (Teubner et al., 2019).*

Besides, the authors separated GPP into Ra and NPP components, so the model can provide the simulations of Ra (as least maintenance respiration) and NPP respectively, which may help us to

judge if the structure of model is reliable. However, the manuscript also missed this kind important information.

Response (6): The separation of GPP into Ra and NPP in Eq. 3.1 is done to introduce the theoretical basis of the VOD2GPP model and is intended to support the theoretical understanding of our modelling approach. The VOD2GPP model itself is implemented using a data-driven machine learning model (GAM) which we trained with the input variables against in-situ GPP observations. Thus, the current model formulation is indeed not constructed to separate GPP into NPP and Ra. But it has the potential to provide the basis for such a modelling approach of explicitly representing Ra and NPP components in the future.

Another important model flaw, if the changes of biomass is equal to GPP. Obviously, the answer is NO! When the biomass lost partly at the 0.25 degree pixel, the GPP should be positive assuming the rest vegetation in this pixel still live, but lost biomass may larger than actual GPP contributed by the rest vegetation.

Response (7): We would like to point out that changes in biomass are not considered equal to GPP in our model. GPP is represented as the sum of both terms, Ra and NPP, with biomass contributing to the maintenance part of Ra and changes in biomass relating to the growth part of Ra as well as to NPP. For further considerations regarding biomass changes, please also see our Response (2) above.

Then, VODCA2GPP product may produce negative GPP.

Response (8): Our modelling approach produces almost entirely positive estimates of GPP. In some very rare cases (2.5% of all data points) we obtained GPP that is slightly negative. However, this is related to the extrapolation capacities of the trained GAM model and is common to most machine learning-based GPP estimates (also FLUXCOM includes a few negative values because of invalid extrapolations). We set all negative GPP values to zero in the final dataset.

We added a more detailed explanation for the mentioned negative GPP:

*Page 8, line 239: "In rare cases (~2.5% of all data points) the machine learning model produced slightly negative results for GPP, due to the extrapolation capacities of the trained GAM. As negative GPP is not possible, such estimates were set to zero."*

Third, as the dataset description paper, I majorly pay much more attention to validation of dataset. The authors actually conducted very weak model validations, only validate the performance for reproducing spatial variations of GPP (low performance as the above comment). I would like to see if the product can reproduce the seasonal and interannual variations of GPP against the measurements of eddy covariance towers.

Response (9): Regarding our model validations we understand the concern that large parts of the validations are not based on in-situ GPP from eddy-covariance towers. This is due to the very limited availability of long-term and continuous in-situ observations of GPP (cf. Table B1) which drastically complicates a comprehensive in-situ validation of VODCA2GPP, and all other existing global GPP datasets. Nevertheless, we conducted an analysis of long-term GPP variability where FLUXNET station data availability allowed it (Fig. A6) and analyzed the overall performance using FLUXNET data (Fig 1d, Fig 4, Fig5, Fig A1, Fig A2, Fig A3). For the other validations we relied on state-of-the-art products of which the quality is assumed to be good.

In order to give a more thorough overview on the performance of VODCA2GPP on different time-scales we added several validations focusing on the comparison with in-situ GPP estimates from FLUXNET in the revised manuscript. Specifically, we added the following validations:

- A site-based cross-validation from which we computed the following error metrics: Root mean square error, bias and Pearson's r. This analysis was conducted for different time-scales: 8-daily, monthly and yearly (Fig 5). The adapted methods section (*3.5 Site-level evaluation and uncertainty assessment*) reads as follows:

Page 8, Line 243: *"The robustness of the model was evaluated based on a site-based cross-validation analysis during which the influence of the selection of available in-situ stations on the GPP model was investigated. For the cross-validation 10 VODCA2GPP models were trained. Each of the 10 models was trained with 90% of the available FLUXNET stations, while the remaining 10% of the stations were retained for validation (Teubner et al., 2019). Every station was excluded exactly once which is why this approach is classified as pseudo-random. Model performance was assessed at all sites that were omitted in the respective model run by computing Root Mean Square Errors (RMSE), Bias, and Pearson's r for different time scales (8-daily, monthly and yearly). [...]*

[Figure]

We added the following text in the results section:

Page 13, Line 321: *"The site-based spatial cross-validation also exhibits only a small (negative) bias of VODCA2GPP (Fig. 5) for monthly and 8-daily GPP values while the bias for annual variations is positive and slightly higher. High median Pearson's r for 8-daily and monthly values indicates good model performance for interannual variations. Pearson's r drops substantially at annual scale which is partly explained by the short period covered by most FLUXNET time series (mean observation time span: 7.27 years), and thus may show little interannual variability. The RMSE decreases with increasing observation length scales."*

- Site-level comparisons with eddy covariance measurements for 8-daily (Fig 3) and monthly (Fig A2) sampling (in addition to the mean yearly values that were already included (Fig A1)). In order to allow comparability with the current state-of-the art we conducted these analysis not only for VODCA2GPP but also for MODIS and FLUXCOM.

[Figure]

**Figure 4: GPP from FLUXNET plotted against GPP from VODCA2GPP, MODIS and FLUXCOM for the period 2002-2016 with 8-daily sampling.**

[Figure]

**Figure A2: GPP from FLUXNET plotted against GPP from VODCA2GPP, MODIS and FLUXCOM for the period 2002-2016 with monthly sampling.**

In addition, the authors seem to hide the magnitude of global VODCA2GPP GPP. From fig. 6, VODCA2GPP estimates for global mean GPP intensity is almost twice than that of MODIS. If I am correct, the global GPP derived from MODIS is 108 Pg C yr-1, thus the global magnitude derived from VODCA2GPP may reach to 200 Pg C or larger? This is a new and the maximum estimates of global GPP that I ever knew. The authors need provide enough robust evidences to prove this number is correct, else I can't trust the VODCA2GPP, which will lead us to a wrong way.

Response (10): We do not hide anything: In all our figures it is clearly visible that VODCA2GPP has a positive bias compared to other global GPP datasets. We transparently discussed this in section 5.4 and we also show that the bias is much smaller when comparing VODCA2GPP to eddy-covariance measurements (Fig 1d, Fig 4, Fig5, Fig A1, Fig A2, Fig A3). In fact, for most landcover types VODCA2GPP exhibits only very mild overestimation or even underestimation (Fig A3). It is true that VODCA2GPP derived estimates of yearly GPP reach 200 ± 2.2 Pg C yr-1 which is indeed much higher than yearly GPP derived from MODIS. However, the comparison with MODIS/FLUXCOM GPP alone is not very meaningful as MODIS GPP, as well as FLUXCOM GPP, are known to substantially underestimate global GPP (Turner et al. 2006; Wang et al., 2017). Other studies suggested values that are much closer to those of VODCA2GPP. Welp et al. (2011) for example came to the conclusion that current estimates of GPP are too low and they estimated the actual yearly GPP to be in the range between 150-175 Pg C yr-1. Koffi et al. (2012) came to similar conclusions (146 ± 19 Pg C yr-1). We agree that VODCA2GPP has a tendency for overestimating GPP in very arid regions ((woody) savannas and open shrublands; Fig A3) but given the high uncertainties for global annual GPP

estimates and the missing consensus regarding this value among the existing datasets we do not see this as an inherent flaw of the VODCA2GPP dataset.

- We added the amount of global annual GPP (200 ± 2.2 Pg C yr-1) in the revised manuscript (Page 11, Line 283).
- In Fig A1. and Fig. A3 (previously Fig. 3 and Fig A1), we adapted the units, replaced the 1:1 line and added the bias in the performance metrics:

[Figure]

**Figure A1: Mean annual in-situ GPP (FLUXNET) plotted against mean annual GPP from VODCA2GPP, FLUXCOM and MODIS for the respective grid cells. Mean annual GPP was computed from all available overlapping years and thus each station is represented by one dot. Red lines indicate the best linear fits determined by ordinary linear regression and the black lines represent the 1:1 lines.**

[Figure]

**Figure A3: Scatterplots of mean annual GPP for FLUXCOM, MODIS and VODCA2GPP for the period 2002-2016 per vegetation type. Vegetation types indicate the pre-dominant IGBP-vegetation type at the respective FLUXNET station.**
**Abbreviations: CRO: Croplands; ENF: Evergreen Needleleaf Forests; DBF: Deciduous Broadleaf Forests; WET: Permanent Wetlands; WSA: Woody Savannas; MF: Mixed Forests; GRA: Grasslands; OSH: Open Shrublands; SAV: Savannas; EBF: Evergreen Broadleaf Forests**

Forth, I felt disappointed and the writing is very poor. There are some low-level mistakes in writing and format. All sections of the manuscript look very rough. Especially, the introduction and discussion are meaningless mostly, and the authors kept some sentences to repeat rather than discussing important scientific questions. Several paragraphs are very short consisting of only 2-3 sentences, which failed to provide useful information.

Feeling so bad when I was reading.

Response (11): We are sorry that the reviewer did not like our style of writing.

We thoroughly revised and improved the manuscript. We rewrote most sections and merged/removed paragraphs when necessary. As our changes concerning the style of writing are

extensive we do not list them here explicitly but we would like to refer you to our submitted "Author's Tracked Changes" version where every improvement is documented in detail.

Some minor comments.
line 49-54: this paragraph is not necessary.

Response (12): We added this paragraph to introduce the most important source of GPP data. We find it necessary to shortly introduce FLUXNET as it is used both for training and validation.

We made the introduction of FLUXNET more concise but we decided to keep this paragraph not only because we believe it is important for the readers but also to appreciate the FLUXNET initiative. The revised paragraph reads as follows:

Page 2, Line 48: *"Locally, GPP can be determined at in-situ flux towers, which measure the net exchange of carbon dioxide by means of eddy-covariances that are partitioned into component GPP and ecosystem respiration fluxes (Baldocchi, 2003). FLUXNET (Pastorello et al., 2020) is the global network of flux towers covering all major biomes, and provides the scientific community with harmonized and well-documented flux observations. FLUXNET stations, however, are sparsely and unevenly distributed, which complicates the derivation of GPP globally."*

It seems the subtitle of "4. Results" missed.

Response (13): Thank you for making us aware of the missing subtitle.

We added the subtitle in the revised manuscript.

There are no GPP values over the desert area in fig. 1c, but there are uncertainties of GPP in fig. 2. These two maps should keep constant for non-vegetated area.

Response (14): The reason why there are no values over some areas in the maps from Fig 1 is that for these comparisons we only used data that are available in all datasets in order to allow comparability between the products. For the uncertainty map we used all data that is available in VODCA2GPP which is why there are also values over the desert.

In order to avoid confusions we added map of mean annual VODCA2GPP (Fig 2a) which is consistent with the uncertainty map (Fig 2b):

[Figure]

**Figure 2: a) Mean annual GPP as derived from VODCA2GPP for the period 1988-2019. b) Standard deviation of mean annual GPP (1988-2019) as obtained by the uncertainty analysis based on site-level cross-validation.**

Fig. 3 shows the poor performance of VODCA2GPP compared to fluxcom and modis products. So why do we need VODCA2GPP product?

Response (15): As explained above, Fig 3 (now: Fig A1) indeed shows lower performance of VODCA2GPP compared to FLUXCOM and MODIS. However, Fig 3 should be analyzed in combination with Fig. A1 which shows that there are certain landcovers that substantially drive the lower performance while others perform well. Keeping these differences in mind, VODCA2GPP can be seen as a reliable data source. Furthermore, VODCA2GPP utilizes microwave remote sensing for predicting GPP and is thus largely independent from the mentioned optical remote sensing based datasets. The advantage of this independence is, for example, visible in the long-term trend analysis (Fig. 7) where some trends are captured by VODCA2GPP which are also visible in TRENDY but not in MODIS or FLUXCOM. Also, the comparison of global anomalies (Fig 6, Table 2) suggests that VODCA2GPP is able to give additional insights in GPP that might be hidden in MODIS or FLUXCOM. Apart from these advantages, VODCA2GPP is also not prone to cloud cover which enables the continuous monitoring of frequently clouded regions like the tropics.

We added section 5.4 where we discuss potential applications of VODCA2GPP and highlight that VODCA2GPP, just like any other global GPP dataset, cannot be seen as "true reference" but can help to gain insights that might be hidden in other datasets:

*Page 20, Line 465: "***5.4 Potential applications of VODCA2GPP***

*The validation results show that VODCA2GPP generally exhibits a high consistency with in-situ GPP observations and global state-of-the art GPP products indicating that VODCA2GPP can be used complementary to current global GPP products. For the analysis of global as well as regional GPP anomalies, VODCA2GPP can provide valuable insights that might be hidden in other observational products due to the fundamentally differing observation methods and the associated limitations related to saturation effects, cloud cover, and other atmospheric effects such as water vapour content or aerosols (Xiao et al., 2019).*

*Also, for the monitoring of global GPP trends, VODCA2GPP has the potential to serve as independent and reliable source of data. The long-term trend analysis suggests that the majority of biomes have increased their primary productivity since 1988. There are several potential drivers for long-term increases in GPP, the most important ones being global warming, land-use changes and elevated CO2 concentration in the atmosphere (Piao et al., 2019). The observed long-term trends in GPP across the different products support the theory of elevated atmospheric CO2 leading to an increased uptake of CO2 (Haverd et. al, 2020; Walker et al., 2020; Campbell et al., 2017; Schimel et al., 2015). The absence of trends in FLUXCOM does not contradict but rather supports this theory as FLUXCOM does not account for CO2 fertilization effects (Jung et al., 2020). Due to the shortness of most in-situ GPP time series, it is, however, difficult to draw final conclusions on the existence, magnitude and reasons for long-term variations of GPP. Therefore, the global influence of atmospheric CO2 on vegetation productivity remains uncertain, but VODCA2GPP allows to gain new perspectives on long-term GPP trends and might help to identify and quantify driving factors for increasing long-term primary productivity.*

*Furthermore, VODCA2GPP can be used as a largely independent source of data for the intercomparison and validation of other existing or newly developed global GPP datasets and models. Currently, there exists a multitude of global GPP products and there exist large inconsistencies among*

*products (Zhang and Aizhong, 2021). Similar to other global GPP dataset, VODCA2GPP cannot be seen as a true reference but using it as additional reference might help to acquire a more comprehensive picture on the performance of other datasets especially in context of long- and short-term variability in GPP."*

4.2 section. The authors only compared temporal variability of VODCA2GPP with modis and fluxcom datasets. How about VODCA2GPP to reproduce the interannual variability of GPP compared to fluxnet observations. Actually, the comparison with fluxnet observations is more important for this data description paper to let the readers and potential users to know if VODCA2GPP can reproduce the long-term or interannual variability of GPP.

Response (16):

We added a cross-validation analysis for VODCA2GPP vs. FLUXNET and included further comparisons with eddy covariance data on different time scales. Please find our improvements regarding the validations in response 9. Additionally to those, we updated Fig A6 and included the exact magnitude (incl. upper and lower confidence intervals) of the derived long-term trends:

[Figure]

**Figure A6: Exemplary collection of time series of in-situ FLUXNET GPP together with extracted time series from MODIS, FLUXCOM and VODCA2GPP. The stations were selected because of their high data availability for the respective landcover. The lines indicate the regression lines as obtained from the Theil-Sen slope estimation for yearly median GPP. The trends are computed for the common observation period with FLUXNET. The slope [g C m-2 yr-1] is depicted in the legend together with the respective 90% lower/upper confidence intervals. The depicted stations are: a) AU-Tum: Tumbarumba, Australia; Lat: -35.65 °N, Lon: 148.15 °E; Landcover: EBF b) DE-Tha: Tharandt, Germany; Lat: 50.96 °N, Lon: 13.57 °E; Landcover: ENF c) GF-Guy: Guyaflux, French Guiana; Lat: 5.28 °N, Lon: -52.93 °E; Landcover: EBF d)US-Ha1: Harvard Forest EMS Tower, United States; Lat: 42.54 °N, Lon: -72.17 °E; Landcover: DBF**

4.3 section. Why did the authors add the trendy simulations here, not compared previously?

Response (17):

We made this clearer in the revised manuscript:

*Page 3, Line 93:" In our analysis, we compare the VODCA2GPP dataset mainly with other data-driven products (FLUXNET, MODIS and FLUXCOM). However, FLUXCOM does not account for CO2 fertilization effects (Walker et al, 2020) which is why trends derived from FLUXCOM are not realistic (Jung et al., 2020). Therefore, we assess monthly anomalies and long-term trends in VODCA2GPP also against TRENDY models, which consider CO2 fertilization."*

4.4 section. Totally confused. Sometimes, the authors made comparison from 2002 to 2016, and otherwise 2003-2015? Can not understand the purpose of the authors.

Response (18): We agree that this is confusing and we apologize for making this not clear enough in the manuscript. The reason why the observation periods differ slightly is the fact that for the used MODIS and FLUXCOM data the observations only start in mid-January 2002 and end in mid-December 2016. Since for the long-term trend analysis we used the yearly median to compute Theil-Slope, we could not assure that these missing observations do not impact the results of the trends. Therefore, we decided to exclude the first and last year in these computations. For the other comparison (e.g. Fig 1), single missing observation do not have any noticeable influence on the metrics which is why we decided to use the fully available periods between 2002 and 2016.

We clarified this in the revised manuscript by updating the description in Table 3:

Page 16, Line 372: *"Table 3: […] The analysed periods are 2003-2015 which corresponds to the fully overlapping period for all datasets (for the period 2002-2016 there were some data gaps in the used MODIS and FLUXCOM data at the very beginning/very end. Since these data gaps could potentially impact the slope estimation, the slightly shorter period 2003-2015 was used) […]"*

Line 396-399: can't accept this explanation for poor model performance. The sparse observations in arid and semi-arid regions will reduce the performance of modis and fluxcom, but VODCA2GPP showed the obvious bias.

Response (19): This explanation is of course not the only reason for uncertainties in VODCA2GPP but, as for all data-driven GPP products, the limited availability of in-situ GPP observations is known to decrease model performance, also for MODIS and FLUXCOM. Other, VODCA2GPP specific reasons for uncertainties in our product are discussed in the sections 5.1 and 5.3.

We added various validations to give a more thorough overview on the performance of VODCA2GPP vs. FLUXNET (please see Response 9). We revised the discussion (Sect. 5.3) by including the insights gained from these analysis:

Page 19, Line 435: *"There is only a very small bias when comparing VODCA2GPP with eddy-covariance measurements from FLUXNET but relatively large discrepancies in absolute GPP exist between VODCA2GPP and other remote-sensing-based products, and to a lesser extent with process-based TRENDY models. […]"*

5.2 section. What is the objective of this section? I did not get any meaning points. The manuscript seems to be rough at the initial stage, and the authors need much more work to improve the writing.

Response (20): In this section, we want to make the reader aware that validations of global GPP datasets, also of VODCA2GPP, are difficult due to the uneven distribution and generally low availability of eddy-covariance measurement sites (see also Jung et al. (2020)). Furthermore, the

objective of this section is to point out that since MODIS and FLUXCOM also heavily rely on FLUXNET data, they cannot be viewed as fully independent data sources although they are using fundamentally differing modelling approaches. Therefore, this section is primarily meant to transparently inform the reader about a potential but unavoidable bias in the validations with FLUXCOM and MODIS due to the partly shared data sources.

We removed this section in the revised manuscript and instead rewrote this paragraph and included it in Sect. 5.3.

5.5 section. I can't agree to this point. There are many factors may enhance the global gpp, how the author can claim the increasing trend imply the CO2 fertilization? There are many conclusions that the authors reached without robust evidences. By the way, this section is basically repeat of results.

Response (21): As we already wrote "there are several potential drivers for long-term increases in GPP", but various studies concluded that CO2 fertilization effects are probably the most important drivers for increasing GPP (e.g. Haverd et. al, 2020; Walker et al., 2020; Campbell et al., 2017; Schimel et al., 2015). We are clearly stating that our results cannot be seen as evidence for this specific effect.

We rewrote this section (now Sect 5.4) into a section which gives an overview on potential applications of VODCA2GPP. The derivation and comparison of trends is one potential application. Possible reasons for trends in GPP are still discussed but we once more highlighted that our results cannot be seen as robust evidence. Please see response 15 for the exact changes.

5.6 section. Again, a meaningless paragraph and section. It looks like a conclusion or summary

Response (22): We do not agree that this section is meaningless but the reviewer is right that parts of this section would indeed fit better in section 7 (Conclusion).

We removed this section in the revised manuscript and instead included the information in other parts of the manuscript where it is better suited (mainly in Sect. 7 Conclusion):

Page 21, Line 498: *"[...] The analysis of monthly anomalies exhibited various extreme events in VODCA2GPP that are also found in one or more existing product indicating high plausibility of VODCA2GPP derived anomalies. Furthermore, trends derived from VODCA2GPP contain several plausible patterns that match those derived from the TRENDY-v7 simulations but are not visible in both observational products and vice versa. This suggests that the novel microwave-based approach in VODCA2GPP has the potential to reveal novel findings about temporal dynamics in GPP at large scales that are not yet captured by other GPP products."*

References

Campbell, J. E., Berry, J. A., Seibt, U., Smith, S. J., Montzka, S. A.,Launois, T., Belviso, S., Bopp, L., and Laine, M.: Large historical growth in global terrestrial gross primary production, Nature, 544, 84–87, , https://doi.org/10.1038/nature22030, 560 2017

Dorigo, W., Wagner, W., Albergel, C., Albrecht, F., Balsamo, G., Brocca, L., Chung, D., Ertl, M., Forkel, M., Gruber, A., Haas, E., Hamer, P. D., Hirschi, M., Ikonen, J., de Jeu, R., Kidd, R., Lahoz, W., Liu, Y. Y., Miralles, D., Mistelbauer, T., Nicolai-Shaw, N., Parinussa, R., Pratola, C., Reimer, C., van der Schalie, R., Seneviratne, S. I., Smolander, T., and Lecomte, P.: ESA CCI Soil Moisture for improved Earth system understanding: State-of-the art and future directions, Remote Sensing of Environment, 203, 185–215, https://doi.org/10.1016/j.rse.2017.07.001, 2017.

Frappart, F., Wigneron, J. P., Li, X., Liu, X., Al-Yaari, A., Fan, L., ... & Baghdadi, N. (2020). Global monitoring of the vegetation dynamics from the Vegetation Optical Depth (VOD): A review. Remote Sensing, 12(18), 2915.

Haverd, V., Smith, B., Canadell, J. G., Cuntz, M., Mikaloff-Fletcher, S., Farquhar, G., Woodgate, W., Briggs, P. R., and Trudinger, C. M. Higher than expected co2 fertilization inferred from leaf to global observations. Glob Change Biol, 26(4):2390–2402, https://doi.org/10.1111/gcb.14950, 2020.

Jung, M., Schwalm, C., Migliavacca, M., Walther, S., Camps-Valls, G., Koirala, S., Anthoni, P., Besnard, S., Bodesheim, P., Carvalhais, N., Chevallier, F., Gans, F., Goll, D. S., Haverd, V., Köhler, P., Ichii, K., Jain, A. K., Liu, J., Lombardozzi, D., Nabel, J. E. M. S., Nelson, J. A., O'Sullivan, M., Pallandt, M., Papale, D., Peters, W., Pongratz, J., Rödenbeck, C., Sitch, S., Tramontana, G., Walker, A., Weber, U., and Reichstein, M.: Scaling carbon fluxes from eddy covariance sites to globe: synthesis and evaluation of the FLUXCOM approach, Biogeosciences, 17, 1343–1365, https://doi.org/10.5194/bg-17-1343- 620 2020, 2020.

Koffi, E. N., Rayner, P. J., Scholze, M., and Beer, C. (2012), Atmospheric constraints on gross primary productivity and net ecosystem productivity: Results from a carbon-cycle data assimilation system, Global Biogeochem. Cycles, 26, GB1024, doi:10.1029/2010GB003900.

Li, X., Wigneron, J. P., Frappart, F., Fan, L., Ciais, P., Fensholt, R., ... & Moisy, C. (2021). Global-scale assessment and inter-comparison of recently developed/reprocessed microwave satellite vegetation optical depth products. Remote Sensing of Environment, 253, 112208.

Schimel, D., Stephens, B. B., and Fisher, J. B. Effect of increasing co2 on the terrestrial carbon cycle. P Natl Acad Sci, 112(2):436–441, https://doi.org/10.1073/pnas.1407302112, 2015.

Teubner, I. E., Forkel, M., Jung, M., Liu, Y. Y., Miralles, D. G., Parinussa, R., van der Schalie, R., Vreugdenhil, M., Schwalm, C. R., Tramontana, G., Camps-Valls, G., and Dorigo, W. A.: Assessing the relationship between microwave vegetation optical 725 depth and gross primary production, Int. J. Appl. Earth Obs., 65, 79–91, https://doi.org/10.1016/j.jag.2017.10.006, 2018.

Teubner, I. E., Forkel, M., Camps-Valls, G., Jung, M., Miralles, D. G., Tramontana, G., van der Schalie, R., Vreugdenhil, M., Mösinger, L., and Dorigo, W. A.: A carbon sink-driven approach to estimate gross primary production from microwave satellite observations, Remote Sens. Environ., 229, 100–113, https://doi.org/10.1016/j.rse.2019.04.022, 2019.730

Teubner, I. E., Forkel, M., Wild, B., Mösinger, L., and Dorigo, W.: Impact of temperature and water availability on microwave-derived gross primary production, Biogeosciences, 18, 3285–3308, https://doi.org/10.5194/bg-18-3285-2021, 2021.

Turner, D. P., Ritts, W. D., Cohen, W. B., Gower, S. T., Running, S. W., Zhao M., Costa, M. H., Kirschbaum, A. A., Ham, J. M., Saleska, S. R., and Ahl, D. E.: Evaluation of MODIS NPP and GPP products across multiple biomes, Remote Sens. Environ., 102, 282–292, https://doi.org/10.1016/j.rse.2006.02.017, 2006.

Walker, A. P., De Kauwe, M. G., Bastos, A., Belmecheri, S., Georgiou, K., Keeling, R., McMahon, S. M., Medlyn, B. E., Moore, D. J. P., … and Zuidema, P. A.: Integrating the evidence for a terrestrial carbon sink caused by increasing atmospheric CO2. New Phytologist, 1. https://doi.org/10.1111/nph.16866, 2020.

Wang, L., Zhu, H., Lin, A., Zou, L., Qin, W., and Du, Q.: Evaluation of the Latest MODIS GPP Products across Multiple 770Biomes Using Global Eddy Covariance Flux Data, Remote Sensing, 9, 5, https://doi.org/10.3390/rs9050418, 2017.

Welp, L. R., Keeling, R. F., Meijer, H. A., Bollenbacher, A. F., Piper, S. C., Yoshimura,K., Francey, R. J., Allison, C. E., and Wahlen, M.:Interannual variability in the oxygen isotopes of atmospheric CO2 driven by El Niño. Nature, 477(7366):579–582, https://doi.org/10.1038/nature10421, 2011.

Woodhouse, I.H. (2005). Introduction to Microwave Remote Sensing. CRC Press.

This study introduced a new global GPP dataset using satellite-derived vegetation optical depth (VOD) dataset spanning 1988-2020 in spatial resolution of 0.25o. The method for relating GPP to VOD was proposed in Teubner et al (2019, 2021). This study also presented validation against FLUXNET observation and other global GPP datasets (MODIS, FLUXCOM, TRENDY models). The difference of spatial and temporal GPP dynamics from other GPP reference datasets were shown. However, given the lower VOD-GPP model performance and extremely high global amount of VODCA2GPP, I do have concern about the quality of this dataset and its advantage over other datasets. The writing also needs a lot of improvements since there are both phrasing mistakes and technical mistakes.

*Please note: All references to materials (e.g. Figures, Tables, Sections) now refer to the numbering of the revised manuscript.*

Response: Dear Referee,

Thank you very much for reviewing our paper and giving very constructive feedback. We understand the concerns regarding the quality of the VODCA2GPP dataset but we strongly believe that although VODCA2GPP shows a slightly weaker performance than current state-of-the-art GPP products when comparing mean annual GPP, VODCA2GPP can still be a very valuable data source for studying the global productivity.

It is true that VODCA2GPP GPP estimates are relatively high compared to estimates from current state-of-the-art products. On the other hand, MODIS and FLUXCOM are known to substantially underestimate GPP especially in very productive regions (Fig 4, Fig A1/A2/A3; Turner et al. 2006; Wang et al., 2017) which is why large differences between MODIS/FLUXCOM and VODCA2GPP do not necessarily implicate substantial overestimation of VODCA2GPP. In fact, Welp at al. (2011) suggested that current estimates of GPP are generally too low and they estimated the actual yearly GPP to be between 150-175 Pg C yr$^{-1}$. Koffi et al. (2012) came to similar conclusions (146 ± 19 Pg C yr$^{-1}$). The comparison of mean annual GPP between VODCA2GPP and FLUXNET GPP does not suggest an overall overestimation of VODCA2GPP but exhibits lowest bias for VODCA2GPP among all analyzed datasets (e.g. Fig 4). Nevertheless, we agree that VODCA2GPP has a tendency for overestimation for arid regions (e.g., (woody) savannas and open shrubland; Fig A3). However, due to the missing consensus among existing datasets (Anav et al., 2015) we do not see this as an inherent flaw of VODCA2GPP but rather as a systematic problem in the observation of GPP originating from generally low in-situ data availability.

We improved the style of writing by rephrasing large parts of the manuscript and we removed technical mistakes. Additionally to the changes listed below we would like to refer you also to the

"Author's Tracked Changes" version of our manuscript where every improvement is documented in detail.

Specific comments:

1 The global total amount of VODCA2GPP datasets is not given.

The global total GPP as derived from VODCA2GPP amounts on average to 200 ± 2.2 Pg C yr$^{-1}$.

We added this number in a revised version of the manuscript:
Page 11, Line 282: *"The mean global total GPP as derived from VODCA2GPP amounts to 200 ± 2.2 Pg C yr-1."*

2 line 19, line31, it should be 'at global scale'.

Thank you for this suggestion. We have revised this in the latest version:

Page 1, Line 18: *"Here, we introduce the new VODCA2GPP dataset, which utilizes microwave remote sensing estimates of Vegetation Optical Depth (VOD) to estimate GPP at global scale for the period 1988 - 2020."*

Page 1, Line 31: *"A trend analysis for the period 1988-2019 did not exhibit a significant trend in VODCA2GPP at global scale but rather suggests regionally different long-term changes in GPP."*

3 line 32, it should be 'different'.

Thank you for this suggestion. We have revised this in the latest version:

Page 1, Line 31: *"A trend analysis for the period 1988-2019 did not exhibit a significant trend in VODCA2GPP at global scale but rather suggests regionally different long-term changes in GPP."*

4 line 32-33, it should be 'similar significant increase'. Please check the usage of two 'for'.

Thank you for this suggestion. We have revised this in the latest version.

Page 1, Line 32: *"For the shorter overlapping observation period (2003-2015) of VODCA2GPP, MODIS GPP, and the TRENDY-v7 ensemble significant increases of global GPP were found."*

*5 line 43-44, please reformulate it.*

Thank you for this suggestion. We have revised this in the latest version:

Page 2, Line 41: *"GPP therefore plays a key role in mitigating the negative effects of anthropogenic emissions."*

6 line 46-47, please reformulate it.

Thank you for this suggestion. We have revised this in the latest version:

Page 2, Line 44: *"Quantifying GPP is essential to understand the effect of climate variability and changes in atmospheric CO2 concentrations on the land carbon cycle (e.g., Baldocchi et al., 2016; Nemani et al., 2003)."*

7 line 51, please use clause.

Thank you for this suggestion. We have revised this in the latest version:

*Page 2, Line 48: "Locally, GPP can be determined at in-situ flux towers, which measure the net exchange of carbon dioxide by means of eddy-covariances that are partitioned into component GPP and ecosystem respiration fluxes (Baldocchi, 2003). FLUXNET (Pastorello et al., 2020) is the global network of flux towers covering all major biomes, and provides the scientific community with harmonized and well-documented flux observations."*

8 line 53, what did you mean by 'measurement bias'? Is it affected by the distribution of flux towers or just measurement techniques?

Here, we refer to the uneven distribution of flux towers which leads to the under/overrepresentation of certain landcovers/biomes in the training data.

We removed this from the introduction and instead left the note on the underrepresentation of certain biomes in the FLUXNET data in the discussion where it is more relevant:

Page 18, Line 408: "[…] *the observation bias which is introduced by unevenly distributed FLUXNET sites decreases the model's robustness. GPP is measured in situ only at a few locations and these stations are mostly located in temperate regions (e.g., Europe and North America) while semi-arid and tropical forest regions are underrepresented in the training data."*

9 line 236, please reformulate it.

Thank you for this suggestion. We have revised this in the latest version:

Page 8, Line 239: *In rare cases (~2.5% of all data points) the machine learning model produced slightly negative results for GPP, due to the extrapolation capacities of the trained GAM. As negative GPP is not possible, such estimates were set to zero.*

10 line 245, for site-level validation, you should also use cross validation method.

We included a site-based cross-validation in the revised manuscript. Please see (13) for the results of the cross-validation and all other improvements regarding the validation of VODCA2GPP.

11 line 264, the subtitle for section 4 is missing.

Thank you for making us aware of the missing subtitle.

We have added the subtitle in the latest version (Page 10, Line 268)

12 In figure 2, the uncertainty is very high in desert area (more than ten time of the GPP itself, why?), for example, northern Africa, which is blank in figure 1. Please use single-hue color series and check to ensure the consistency with figure 1.

Unfortunately, the scaling of the uncertainty plot (Fig 2b) was not correct.
We adapted the scaling in the revised manuscript (Fig 2b) and apologize for this mistake.

Nevertheless, it is true that the uncertainties are very high in some very arid regions such as the Sahara. The exact reason for the low robustness of the model in these regions is unclear but we assume that these instabilities are partly explicable with missing training data from desert areas.

The reason for the missing consistency between Fig 1a and Fig 2b is that in Fig 1a we only used data that is available in all three products (MODIS, FLUXCOM, VODCA2GPP) in order to allow comparability.

We agree that this is confusing and added an additional figure (Fig 2a) which now includes all available VODCA2GPP data which is consistent with Figure 2b. Furthermore, we updated Figure 2b by using a single-hue color scale and merged it with Fig 2a.

[Figure]

**Figure 2: a) Mean annual GPP as derived from VODCA2GPP for the period 1988-2019. b) Standard deviation of mean annual GPP (1988-2019) as obtained by the uncertainty analysis based on site-level cross-validation.**

13 In figure 3, please check the unit in both x and y axis. Please use dashed line for 1:1 line. Here, you presented the site-level evaluation for mean annual GPP. What about the annual, seasonal (or monthly) evaluation? In fact, Pearson value of around 0.5 seems to be relatively lower for GPP estimation. Did you try 10-fold cross validation for model performance?

VODCA2GPP underperforms mainly for sparsely vegetated biomes, while performance is good for densely vegetated regions (Fig A3). In these regions VODCA2GPP's performance metrics (Pearson's r and RMSE) are comparable or even better than those from MODIS and FLUXCOM. Therefore, we suggest that VODCA2GPP can be used as a reliable source of GPP data as long as the users are aware of the performance differences depending on the landcover. We understand that this can be seen as drawback but considering the fact that VODCA2GPP performs well for most regions and is largely independent from current state-of-the-art products, we believe that VODCA2GPP can provide very valuable input for the scientific community. Due to the independence of VODCA2GPP, it also allows users to gain insights on global GPP trends and anomalies that are partly hidden in optical remote sensing based GPP datasets.

In Fig A1 (previously Fig. 3), we adapted the units, replaced the 1:1 line and added the bias in the performance metrics:

[Figure]

**Figure A1: Mean annual in-situ GPP (FLUXNET) plotted against mean annual GPP from VODCA2GPP, FLUXCOM and MODIS for the respective grid cells. Mean annual GPP was computed from all available overlapping years and thus each station is represented by one dot. Red lines indicate the best linear fits determined by ordinary linear regression and the black lines represent the 1:1 lines.**

In order to give a more thorough overview on the performance of VODCA2GPP on different time-scales we added several validations focusing on the comparison with in-situ GPP estimates from FLUXNET on different time-scales in the revised manuscript. Specifically, we added the following validations:

- Site-level comparisons with eddy covariance measurements for 8-daily (Fig 3) and monthly (Fig A2) sampling (in addition to the mean yearly values that were already included (Fig A1)). In order to allow comparability with the current state-of-the art we conducted these analysis not only for VODCA2GPP but also for MODIS and FLUXCOM. The

[Figure]

**Figure 4: GPP from FLUXNET plotted against GPP from VODCA2GPP, MODIS and FLUXCOM for the period 2002-2016 with 8-daily sampling.**

[Figure]

**Figure A2: GPP from FLUXNET plotted against GPP from VODCA2GPP, MODIS and FLUXCOM for the period 2002-2016 with monthly sampling.**

- A site-based cross-validation from which we computed the following error metrics: Root mean square error, bias and Pearson's r. This analysis was conducted for different time-scales: 8-daily, monthly and yearly (Fig 5). The adapted methods section (*3.5 Site-level evaluation and uncertainty assessment*) reads as follows:

    Page 8, Line 243: *"The robustness of the model was evaluated based on a site-based cross-validation analysis during which the influence of the selection of available in-situ stations on the GPP model was investigated. For the cross-validation 10 VODCA2GPP models were trained. Each of the 10 models was trained with 90% of the available FLUXNET stations, while*

*the remaining 10% of the stations were retained for validation (Teubner et al., 2019). Every station was excluded exactly once which is why this approach is classified as pseudo-random. Model performance was assessed at all sites that were omitted in the respective model run by computing Root Mean Square Errors (RMSE), Bias, and Pearson's r for different time scales (8-daily, monthly and yearly). [...]*

[Figure]

**Figure 5: Site-based cross-validation for 8-daily, monthly and yearly sampling of VODCA2GPP and FLUXNET GPP. RMSE, Bias and Pearson's r were computed at each of the 10% of FLUXNET sites that were omitted during the respective training run. The whiskers of the boxplots extend to the 0.05/0.95 percentiles.**

We added the following text concerning the cross-validation in the result section:

Page 13, Line 320: *"The site-based spatial cross-validation also exhibits only a small (negative) bias of VODCA2GPP (Fig. 5) for monthly and 8-daily GPP values while the bias for annual variations is positive and slightly higher. High median Pearson's r for 8-daily and monthly values indicates good model performance for interannual variations. Pearson's r drops substantially at annual scale which is partly explained by the short period covered by most FLUXNET time series (mean observation time span: 7.27 years), and thus may show little interannual variability. The RMSE decreases with increasing observation length scales."*

- Structure-wise we merged Sect. 4.2 (Comparison of temporal dynamics) with Sect. 4.1 (Spatial patterns in global annual GPP) into "4.1 Spatiotemporal patterns in global GPP". We added Sect. 4.2 (Site-level evaluation) which includes the above mentioned analysis.

- We updated Fig A6 and included the exact magnitude (incl. upper and lower confidence intervals) of the derived long-term trends:

[Figure]

**Figure A6: Exemplary collection of time series of in-situ FLUXNET GPP together with extracted time series from MODIS, FLUXCOM and VODCA2GPP. The stations were selected because of their high data availability for the respective landcover. The lines indicate the regression lines as obtained from the Theil-Sen slope estimation for yearly median GPP. The trends are computed for the common observation period with FLUXNET. The slope [g C m-2 yr-1] is depicted in the legend together with the respective 90% lower/upper confidence intervals. The depicted stations are:**
**a) AU-Tum: Tumbarumba, Australia; Lat: -35.65 °N, Lon: 148.15 °E; Landcover: EBF**
**b) DE-Tha: Tharandt, Germany; Lat: 50.96 °N, Lon: 13.57 °E; Landcover: ENF**
**c) GF-Guy: Guyaflux, French Guiana; Lat: 5.28 °N, Lon: -52.93 °E; Landcover: EBF**
**d)US-Ha1: Harvard Forest EMS Tower, United States; Lat: 42.54 °N, Lon: -72.17 °E; Landcover: DBF**

14 line 308, when you mention 'good temporal agreement', you need to specify the region, since the correlation is even negative in tropical America.

We agree that this needed to be reformulated more precisely:

Page 12, Line 298: *"VODCA2GPP shows good temporal agreement with MODIS and FLUXCOM for most regions outside the tropics (Fig. 3)."*

15 line 308-309, it should be 'the highest' and 'the lowest'.

Thank you for this suggestion. We have revised this in the latest version.

Page 12, Line 299: *"Pearson's r is the highest in regions with distinct interannual variability such as sub-arctic, temperate, and semi-arid regions and the lowest for dense tropical forests where even negative correlations occur."*

16 line 311, please rephrase it.

Thank you for this suggestion. We have revised this in the latest version.

Page 12, Line 300: *Median Pearson's r between VODCA2GPP and GPP from MODIS and FLUXCOM is 0.77 and 0.75, respectively.*

17 line 313, why monthly GPP anomalies are compared? Why you did not use monthly GPP instead?

Here, we wanted to evaluate VODCA2GPP's ability to capture extreme events and anomalies in general.

18 line 334, here you used 2002-2016. Why did you use 2003-2015 in next paragraph?

Thank you for bringing this to our attention. The period 2003-2015 is correct. We have revised this.

In order to avoid confusions regarding the time period used in the trend analysis of all products we added the following text:

Page 16, Line 372: *"Table 3: […] The analysed periods are 2003-2015 which corresponds to the fully overlapping period for all datasets (for the period 2002-2016 there were some data gaps in the used MODIS and FLUXCOM data at the very beginning/very end. Since these data gaps could potentially impact the slope estimation, the slightly shorter period 2003-2015 was used) […]"*

19 line 335, there are only three sentences in this paragraph. It would be better to merge it with next paragraph.

Thank you for this suggestion. We merged these paragraphs. (Page 15, Line 348)

20 In figure 6, at global scale, the VODCA2GPP is two times of the MODIS and FLUXCOM. So the global GPP can be more than 200 PgC yr-1, which is the highest estimation as far as I know. You should give reasonable evidence for such value, or it won't be convincing.

It is true that VODCA2GPP estimates are substantially higher than MODIS and FLUXCOM estimates but as already outlined above, optical products are known to significantly underestimate GPP, especially in very productive regions (Fig 4, Fig A1/A2/Fig A3; Turner et al., 2006; Wang et al., 2017). Thus, differences in global annual GPP between VODCA2GPP and MODIS/FLUXCOM need to be interpreted with caution. In fact, the comparison of VODCA2GPP with in-situ GPP does not suggest a systematic overestimation. VODCA2GPP only shows strong overestimation for (woody) savannas and open shrublands (Fig A3). For all FLUXNET sites located in other landcovers our analysis suggests very mild overestimation or even underestimation of VODCA2GPP. Furthermore, the comparison of VODCA2GPP with the mean of all TRENDY models (Fig. 7) indicates that the overestimation is by far not as severe as implied by MODIS and FLUXCOM.

We added more validations with in situ GPP from FLUXNET which provide a thorough overview on the performance of VODCA2GPP on different time-scales and prove its overall reliability. Please see (13) for details.

21 In figure 7, it should be 2002-2016 or 2003-2015?

2003-2015 is correct. Please see (18) for more details.

22 In section 5.2, TRENDY ensemble is indeed independent from VODCA2GPP. But I couldn't get the meaning of this paragraph. The process-based estimation without accounting for nutrient constraints is also reported to overestimate GPP.

The objective of this section is to emphasize that the validation of global GPP datasets generally needs to be interpreted with caution due to a) the generally low availability and uneven distribution of FLUXNET stations and b) due to the often shared in-situ data-source (i.e. FLUXNET). MODIS, FLUXCOM and VODCA2GPP use the same reference measurements for calibration and validation. As a result MODIS and FLUXCOM cannot be viewed as completely independent from VODCA2GPP although they are using fundamentally differing modelling approaches. Thus, this paragraph specifically aims to transparently inform potential users about these inevitable validation problems.

We removed this section in the revised manuscript and instead rewrote this paragraph and merged it with Sect. 5.3.

23 In section 5.6, in terms of the VODCA2GPP datasets, more similarities with TRENDY than that of other observational products were mentioned. Is it really a strong point for this GPP dataset?

As already mentioned in response 13, one strong argument for VODCA2GPP is its large independence from other (optical) observational products. This independence is evident especially when analyzing long-term trends in GPP where process-based and optical remote sensing products only exhibit very weak correspondence. VODCA2GPP on the other hand exhibits patterns that are visible in at least one other dataset which led us to the conclusion that VODCA2GPP is able to largely bridge the obvious gap between process-based and optical remote-sensing based products. Thus, not the good correspondence with TRENDY alone is a strong point for VODCA2GPP but more importantly its ability to complement existing products, both process-based and observational, is advantageous.

We removed Sect. 5.6 in the revised manuscript and added the following text in the Conclusion:

Page 21, Line 500: "[…] Furthermore, trends derived from VODCA2GPP contain several plausible patterns that match those derived from the TRENDY-v7 simulations but are not visible in both observational products and vice versa. This suggests that the novel microwave-based approach in VODCA2GPP has the potential to reveal novel findings about temporal dynamics in GPP at large scales that are not yet captured by other GPP products."

24 In figure A1, please check the unit in both x and y axis. You can give annual, seasonal (or monthly) evaluation as well.

We adapted the units in the axis labelling and added the bias in the performance metrics:

[Figure]

**Figure A3: Scatterplots of mean annual GPP for FLUXCOM, MODIS and VODCA2GPP for the period 2002-2016 per vegetation type. Vegetation types indicate the pre-dominant IGBP-vegetation type at the respective FLUXNET station.**
**Abbreviations: CRO: Croplands; ENF: Evergreen Needleleaf Forests; DBF: Deciduous Broadleaf Forests; WET: Permanent Wetlands; WSA: Woody Savannas; MF: Mixed Forests; GRA: Grasslands; OSH: Open Shrublands; SAV: Savannas; EBF: Evergreen Broadleaf Forests**

25 The code for producing the data and figures should also be publicly available in a repository for transparency reason.

We generally embrace the idea of open access science which is why we made the VODCA2GPP dataset publicly and freely available. However, at the moment we do not consider our work finished. We still strive for improvements of the VODA2GPP dataset which we will then of course also share with the scientific (and non-scientific) community. Therefore, we are currently not planning to make the VODCA2GPP code publicly available but we are happy to share the code with anyone who approaches us with ideas for collaboration. Regarding the transparency, we described the model

algorithms and the data that is depicted in the figures in detail which is why we do not see a necessity to explicitly share the underlying code.

References

Anav, A., et al. (2015), Spatiotemporal patterns of terrestrial gross primary production: A review, Rev. Geophys., 53, 785– 818, doi:10.1002/2015RG000483.

Koffi, E. N., Rayner, P. J., Scholze, M., and Beer, C. (2012), Atmospheric constraints on gross primary productivity and net ecosystem productivity: Results from a carbon-cycle data assimilation system, Global Biogeochem. Cycles, 26, GB1024, doi:10.1029/2010GB003900.

Turner, D. P., Ritts, W. D., Cohen, W. B., Gower, S. T., Running,S. W., Zhao M., Costa, M. H., Kirschbaum, A. A., Ham, J. M.,Saleska, S. R., and Ahl, D. E.: Evaluation of MODIS NPP and GPP products across multiple biomes, Remote Sens. Environ.,102, 282–292, https://doi.org/10.1016/j.rse.2006.02.017, 2006.

Wang, L., Zhu, H., Lin, A., Zou, L., Qin, W., and Du, Q.: Evaluation of the Latest MODIS GPP Products across Multiple 770Biomes Using Global Eddy Covariance Flux Data, Remote Sensing, 9, 5, https://doi.org/10.3390/rs9050418, 2017.

Welp, L. R., Keeling, R. F., Meijer, H. A., Bollenbacher, A. F., Piper, S. C., Yoshimura,K., Francey, R. J., Allison, C. E., and Wahlen, M.:Interannual variability in the oxygen isotopes of atmospheric CO2 driven by El Niño. Nature, 477(7366):579–582, https://doi.org/10.1038/nature10421, 2011.

---

## Referee Report (RR1)

Compared to the last version, the manuscript is clearer now and fixes some problems. I still have a few concerns before considering the acceptance.

Minor comments:

1 Why does r drop with increase in time scale but RMSE does the opposite? You can add explanation in the end of line 383.

2 Since the uncertainty metric is derived from the 10 VODCA2GPP models, you can also mention the source of such uncertainty. Is it can be regarded as extrapolation as well?

3 You can also make comparison between your GPP uncertainty and that of other GPP datasets. Indeed, such VOD-GPP dataset is independent of optical-based one, but you really need remind the uncertainty in its application.

4 And a similar question to the last round, the process-based model cannot be treated as ground truth. I don't think the similarity between VODCA2GPP and TRENDY models can be an advantage.

---

## Author Response (AR2)

Formatting as follows:

Reviewer comments
Reply to comments
Changes to the manuscript

**Anonymous Referee (#1)**
Received: 7 November 2021

This is my second round of reviewing. The author's responses confirmed my concerns, and this study or this dataset includes several important flaws which may substantially mislead the future studies in this filed. Therefore, I strongly suggest the authors seriously consider the method and the GPP dataset, and take the efforts to develop the reliable method.

Response: Dear Referee,

Many thanks for reviewing our manuscript again. We appreciate your feedback, and we are sorry that we could not settle your doubts regarding the VODCA2GPP dataset.

First, the authors confirmed their estimate of global GPP reaches to 200 Pg C yr-1, which is almost double of the current estimates. Although the authors argued that their estimates are close to the estimates from Welp et al (2011) (150-175 Pg C yr-1) and Koffi et al. (2012) (146 Pg C yr-1). However, the estimate in this study also is higher than these two studies about higher 30%-50%. Besides, these two studies are based on atmospheric inversion methods to indirectly estimate GPP, and ecosystem respiration may highly impact their estimates to GPP. As the Welp et al (2011) claimed "best guess of 150-175" of GPP. On contrary, MODIS and FLUXCOM used site-based GPP observations to constrain their estimates, and which provide the robust estimates of GPP compared to Welp and Koffi.

Yes, global annual VODCA2GPP reaches 200 Pg C yr-1, which is indeed higher than observation-based GPP derived from MODIS or FLUXCOM. We agree that this positive bias suggests an overestimation of VODCA2GPP at a global scale, and we understand the concerns regarding this overestimation. We devoted a separate section (sect. 5.3) to this issue and discussed the drivers for this positive bias in respect to the optical remote sensing-based references in detail. Thus, we are aware of this issue and transparently inform about it. By no means we claim that the global estimates from our product are closer to the truth than other products.

What we want to highlight again, however, is the fact that when comparing VODCA2GPP with in-situ GPP from FLUXNET we only find substantial overestimation in water-limited regions (e.g., open shrublands and savannas). For most other biomes, the comparison with in-situ GPP does not suggest an overestimation of VODCA2GPP. Furthermore, we want to emphasize that there is no consensus in estimates of global annual GPP among existing datasets (Anav et al. (2015)) which complicates a fair validation of global annual GPP estimates. Thus, we agree that VODCA2GPP derived global annual GPP are too high in some biomes (e.g., arid regions), which presumably also leads to an overestimation at a global scale, but the magnitude of this overestimation cannot be quantified reliably as estimates for global annual GPP are contradictory.

The authors validated their GPP estimates at eddy covariance towers. VODCA2GPP are comparable to tower-based GPP as Fig. 1A shown. I am wondering that there are large differences over the global

estimates. The method may have significant flaw that make it impossible to apply over global scale. Therefore, I strongly suggest the authors investigate the reliability of the method before producing global GPP dataset. As I pointed out that there are several unclear items in the model algorithms, which may induce large uncertainties for GPP estimates. For example, the response #5, the authors changed the definition of mdn(VOD) from landcover to vegetation density. It is totally confused what vegetation density means? By my knowledge, there is no concept of vegetation density, instead that we say Species Density, which is obvious different with the authors' idea. It is my largest concern the authors failed to propose the robust physiological principle for using VOD to estimate GPP at all.

Indeed, VODCA2GPP compares well with in-situ GPP from FLUXNET, which also indicates that VODCA2GPP can in principle be trusted. Also, the underlying method (the VOD2GPP model) is generally reliable, has been peer-reviewed by several experts, and was published in reputable journals (Teubner et al., 2018, 2019, 2021).

As outlined in our revised manuscript, the mdn(VOD) term in the model formulations represents a static vegetation biomass component which helps the model to subtract larger structural vegetation elements (e.g., stems and branches) and thus makes the VOD2GPP model more sensitive to photosynthetically active parts of the vegetation. So to say, the additional term mdn(VOD) allows the model to differentiate between canopy types that have similar VOD dynamics but different above-ground biomass. This is also visible in the partial dependency plot, where we assessed the influence of each input variable on the GPP estimates (Fig. 1 c). Regarding the nomenclature, we believe that vegetation density is a good term for intuitively describing the role of mdn(VOD) in the VOD2GPP model but we are open for other suggestions. As correctly suggested by the reviewer, species density is a different concept.

[Figure]

**Fig 1: Partial dependency plot for GPP and the input variables: VOD (a), $\Delta(VOD)$ (b), mdn(VOD) (c), T2M (d). Dashed lines denote the 95% confidence interval. The interaction term between VOD and T2M is depicted as 3D surface which is bin-wise projected onto a 2D plane for visualization.**

In addition, the authors examined MODIS and FLUXCOM dataset against eddy covariance-based GPP. However, the results in this manuscript look quite different with previous reports. Especially, FLUXCOM is data-driven dataset, which should be compared with site-based GPP. However, the authors showed the underestimated GPP by FLUXCOM, which is quite different with previous studies and also difficulty to understand.

The results of FLUXCOM GPP and MODIS GPP that we present may indeed differ from results presented in other studies, because of a different selection of ground data, and spatial and temporal subsetting. In general, however, our results are in line with what has been reported for FLUXCOM and MODIS GPP (e.g., underestimation of MODIS and FLUXCOM GPP in highly productive regions (Joiner et al., 2018; Anav et al., 2015; Turner et al., 2006)). Nevertheless, differences between our results and those of previous studies can arise for multiple reasons. First, we spatially aggregated

FLUXCOM GPP and MODIS GPP to 0.25° to match VODCA2GPP's resolution. This is most likely not the case for other studies and can thus lead to differences. Furthermore, we only considered pixels that are available in all three datasets (VODCA2GPP, MODIS GPP, and FLUXCOM GPP). Consequently, a certain share of available MODIS/FLUXCOM pixels, which are not available in VODCA2GPP, is not used in our comparisons but is included in other studies. Also, the observed time periods might differ between the studies, which further complicates a direct comparison. Considering these aspects, our results are not directly comparable with results from other studies.

Second, the VODCA2GPP dataset showed the low performance both over spatial and temporal scales. The authors added the validations on model performance for reproducing interannual variability of GPP (response #9). However, the performance is quite low, and mean R2 value is only 0.2 or even lower. By this low performance, I can not trust the capability of VODCA2GPP, and will not use it to conduct any further analyses. So, I still doubted why we still need VODCA2GPP dataset. The authors argued that we need other satellite data source besides optical data, but it is not a reason for accepting its low performance.

We do not share the reviewer's opinion that VODCA2GPP shows an overall low performance. The VODCA2GPP model performs reasonably well for reproducing 8-daily and monthly variations of GPP (Pearson's r: 0.53 and 0.6). It is true that median Pearson's r drops substantially at yearly sampling. This decrease, however, is explicable with extremely low availability of (FLUXNET) data points at this temporal scale. The average time span of FLUXNET time-series are only 7 years. When only considering significant correlations (p-value < 0.1) we find that median Pearson's r reaches much higher values. The number of significant values for Pearson's r at yearly sampling, however, drops to only 8.

We updated Figure 5 by only including significant correlations and we added the number of significant Pearson's r values in the caption to remind the reader about the relatively low expressiveness of this value. Furthermore, we added the following text:

*(Page: 13, Line 326): (…) It is to be noted that there are only 8 significant Pearson's r values for yearly sampling which decreases the expressiveness of this value. This is explicable with the short observation period of most FLUXNET sites which might not exhibit interannual variability. (…)*

[Figure]

**Figure 5: Site-based cross-validation for 8-daily, monthly, and yearly sampling of GPP from VODCA2GPP and FLUXNET. RMSE, Bias and Pearson's r were computed at each of the 10% of FLUXNET sites that were omitted during the respective training run. Non-significant Pearson correlation (p-value < 0.1) were ignored. The boxplots for**

**Pearson's r are based on the 71 (8-daily), 66 (monthly) and 8 (yearly sampling) significant values for Pearson's r values. The whiskers of the boxplots extend to the 0.05/0.95 percentiles.**

**References**

Anav, A., Friedlingstein, P., Beer, C., Ciais, P., Harper, A., Jones,C., Murray-Tortarolo, G., Papale, D., Parazoo, N. C., Peylin, P., Piao, S., Sitch, S., Viovy, N., Wiltshire, A., and Zhao, M.: Spatiotemporal patterns of terrestrial gross primary production: A review, Rev. Geophys., 53, 785–818, doi:10.1002/2015RG000483, 2015.

Joiner, J., Yoshida, Y., Zhang, Y., Duveiller, G., Jung, M., Lyapustin, A., Wang, Y., and Tucker, J. C.: Estimation of Terrestrial Global Gross Primary Production (GPP) with Satellite Data-Driven Models and Eddy Covariance Flux Data, Remote Sens., 10, 1346, https://doi.org/10.3390/rs10091346, 2018.

Teubner, I. E., Forkel, M., Jung, M., Liu, Y. Y., Miralles, D. G., Parinussa, R., van der Schalie, R., Vreugdenhil, M., Schwalm, C. R., Tramontana, G., Camps-Valls, G., and Dorigo, W. A.: Assessing the relationship between microwave vegetation optical 725 depth and gross primary production, Int. J. Appl. Earth Obs., 65, 79–91, https://doi.org/10.1016/j.jag.2017.10.006, 2018.

Teubner, I. E., Forkel, M., Camps-Valls, G., Jung, M., Miralles, D. G., Tramontana, G., van der Schalie, R., Vreugdenhil, M., Mösinger, L., and Dorigo, W. A.: A carbon sink-driven approach to estimate gross primary production from microwave satellite observations, Remote Sens. Environ., 229, 100–113, https://doi.org/10.1016/j.rse.2019.04.022, 2019.730

Teubner, I. E., Forkel, M., Wild, B., Mösinger, L., and Dorigo, W.: Impact of temperature and water availability on microwave-derived gross primary production, Biogeosciences, 18, 3285–3308, https://doi.org/10.5194/bg-18-3285-2021, 2021.

Turner, D. P., Ritts, W. D., Cohen, W. B., Gower, S. T., Running, S. W., Zhao M., Costa, M. H., Kirschbaum, A. A., Ham, J. M., Saleska, S. R., and Ahl, D. E.:  Evaluation of MODIS NPP and GPP products across multiple biomes,  Remote Sens. Environ.,102, 282–292, https://doi.org/10.1016/j.rse.2006.02.017, 2006.

Compared to the last version, the manuscript is clearer now and fixes some problems. I still have a few concerns before considering the acceptance.

Response: Dear Referee,

Thank you very much for reviewing our manuscript again. We appreciate your feedback, and we are happy to address your concerns.

Minor comments:

1 Why does r drop with increase in time scale but RMSE does the opposite? You can add explanation in the end of line 383.

We assume that this comment refers to the cross-validation results depicted in Fig. 5. It is true that Pearson's r drops substantially at yearly sampling. The low Pearson's r values at this timescale, however, are more likely caused by relatively short observation periods of FLUXNET GPP than by weak performance of VODCA2GPP. The average FLUXNET GPP time-series only covers approximately 7 years and thus might only show little to no interannual variability resulting in a high number of non-significant correlations. In fact, the number of significant correlations (p-value < 0.1) is only 8 for yearly sampling. When removing all non-significant values from this analysis, median Pearson's r increases substantially for yearly sampling (from ca. 0.17 to 0.69), while the other time scales (8-daily and monthly) are only slightly affected from the exclusion of non-significant correlations (cf. Fig 5). Excluding non-significant correlations also removes the contrary behavior of Pearson's r and RMSE.

We updated Figure 5 by only including significant correlations and we added the number of significant Pearson's r values in the caption to remind the reader about the relatively low expressiveness of this value. Furthermore, we added the following text:

(Page: 13, Line 326): (…) It is to be noted that there are only 8 significant Pearson's r values for yearly sampling which decreases the expressiveness of this value. This is explicable with the short observation period of most FLUXNET sites which might not exhibit interannual variability. (…)

[Figure]

**Figure 5: Site-based cross-validation for 8-daily, monthly and yearly sampling of GPP from VODCA2GPP and FLUXNET. RMSE, Bias and Pearson's r were computed at each of the 10% of FLUXNET sites that were omitted during the respective training run. Non-significant Pearson correlation (p-value < 0.1) were ignored. The boxplots for Pearson's r are based on the 71 (8-daily), 66 (monthly) and 8 (yearly sampling) significant values for Pearson's r values. The whiskers of the boxplots extend to the 0.05/0.95 percentiles.**

2 Since the uncertainty metric is derived from the 10 VODCA2GPP models, you can also mention the source of such uncertainty. Is it can be regarded as extrapolation as well?

The uncertainty analysis is based on 10 VODCA2GPP models in which 10% percent of the station data was retained during each run. The results of this uncertainty analysis are depicted in Fig 1d and Fig 2b. Fig. 1d suggests that the choice of stations influences the resulting GPP estimates. We find the lowest spread in the 10 models (i.e., lowest uncertainty) north of 20° N where also the majority of FLUXNET GPP stations are located. The Southern hemisphere, where only few in-situ stations are located, generally exhibits a higher spread (larger uncertainty) indicating a considerable sensitivity to the choice of stations. This emphasizes the need for a globally well distributed network of in-situ flux towers.

Thank you for bringing up the extrapolation capabilities of the VODCA2GPP model which were tested through the site-based cross-validation. This analysis revealed that the VODCA2GPP model performs reasonably well at all time scales (Fig 5). The fact that higher correlations are found with global GPP from FLUXCOM and MODIS (median Pearson's r: 0.75 and 077) indicates that FLUXNET stations might not be always representative for the 0.25° VOD pixels.

We added the following text in the revised manuscript:

Page: 11, Line: 286: (…) *The lowest spread in the 10 models (i.e., the lowest uncertainty) is found north of 20°N where also the majority of FLUXNET GPP stations is located. The Southern hemisphere, where only few in-situ stations are located, generally exhibits a larger spread (higher uncertainty) indicating a considerable sensitivity of the model to the choice of stations. This emphasizes the need for a well distributed network of in-situ flux towers across all biomes. (…)*

3 You can also make comparison between your GPP uncertainty and that of other GPP datasets. Indeed, such VOD-GPP dataset is independent of optical-based one, but you really need remind the uncertainty in its application.

Thank you for this suggestion. It is indeed important to also discuss and compare the uncertainties of other global GPP datasets in respect to uncertainties of VODCA2GPP. Uncertainties in optical remote sensing-based GPP are mostly associated with the used wavelength. Optical remote sensing is heavily influenced by weather and illumination conditions. Clouds often contaminate or prevent the observations which is presumably the main reason why the largest uncertainties for FLUXCOM and MODIS are found in the tropics where GPP is known to be underestimated (de Almeida et al., 2018; Jung et al., 2020). In contrast to this, VODCA2GPP shows very good correspondence for densely vegetated areas (e.g., Broadleaf evergreen forests) and is hardly affected by weather conditions. However, we do find comparatively high uncertainties in water-limited areas (e.g., savannas and open shrublands) which presumably originate from multiple sources (which are discussed in the manuscript). The site-based uncertainty analysis also revealed that VODCA2GPP exhibits large uncertainties in mountainous regions with high topographic complexity which has also been reported to decrease reliability of GPP estimates in other GPP products (Xie et al., 2021).

Page: 18, Line: 418: *(…) A comparison of uncertainties between VODCA2GPP and optical remote sensing based GPP (Xie et al., 2021) shows that in both cases topographic complexity decreases the reliability. Furthermore, the reliability of GPP estimates based on optical remote sensing is highly dependent on weather and illumination conditions. Clouds often contaminate or prevent the observations which is presumably the main reason why the largest uncertainties for FLUXCOM and MODIS are found in the wet tropics where GPP is known to be underestimated (de Almeida et al., 2018; Jung et al., 2020). In contrast, VODCA2GPP shows good skill for densely vegetated areas, including broadleaf evergreen forests. On the other hand, the relatively high uncertainties of VODCA2GPP in water-limited regions have not been reported for FLUXCOM or MODIS GPP, indicating that these are VOD-specific and presumably caused by the abovementioned isohydric behavior of plants in arid regions.*

4 And a similar question to the last round, the process-based model cannot be treated as ground truth. I don't think the similarity between VODCA2GPP and TRENDY models can be an advantage.

It is true, that process-based models cannot be treated as "true" reference or "ground truth". The same applies to observation-based estimation approaches such as FLUXCOM or MODIS GPP. The source of data that comes the closest to actual "ground truth" is data from eddy covariance flux measurements. Therefore, our analysis and validations are first and foremost based on the comparison with in-situ estimates from FLUXNET. Due to the sparse and uneven distribution of FLUXNET stations, however, the comparison with other state-of-the-art GPP datasets such as TRENDY GPP is important to assess the validity of GPP estimates at a global scale. In various other GPP related studies, TRENDY GPP has served as reference data set for global patterns in GPP (e.g., O'Sullivan et al., 2020). Especially the evaluation of FLUXCOM GPP at global scale was largely based on the comparison with TRENDY GPP (e.g., Jung et al., 2020). We decided to add TRENDY GPP in our analysis not only to assess the validity of VODCAGPP trends and anomalies at global scale but also to highlight the diversity of GPP estimates that are currently available.

**References**

de Almeida, C. T., Delgado, R. C., Galvao, L. S., de Oliveira Cruz e Aragao, L. E., and Concepcion Ramos, M.: Improvements of the MODIS Gross Primary Productivity model based on a comprehensive uncertainty assessment over the Brazilian Amazonia, ISPRS J. Photogramm. Remote Sens., 145, 268–283, https://doi.org/10.1016/j.isprsjprs.2018.07.016, 2018.

Jung, M., Schwalm, C., Migliavacca, M., Walther, S., Camps-Valls, G., Koirala, S., Anthoni, P., Besnard, S., Bodesheim, P., Carvalhais, N., Chevallier, F., Gans, F., Goll, D. S., Haverd, V., Köhler, P., Ichii, K., Jain, A. K., Liu, J., Lombardozzi, D., Nabel, J. E. M. S., Nelson, J. A., O'Sullivan, M., Pallandt, M., Papale,

D., Peters, W., Pongratz, J., Rödenbeck, C., Sitch, S., Tramontana, G., Walker, A., Weber, U., and Reichstein, M.: Scaling carbon fluxes from eddy covariance sites to globe: synthesis and evaluation of the FLUXCOM approach, Biogeosciences, 17, 1343–1365, https://doi.org/10.5194/bg-17-1343-2020, 2020.

O'Sullivan, M., Smith, W. K., Sitch, S., Friedlingstein, P., Arora, V. K., Haverd, V., Jain, A., Kato, E., Kautz, M., Lombardozzi, D., Nabel, J, Tian, H., Vuichard, N., Wiltshire, A., Zhu, D., and Buermann, W.:Climate-driven variability and trends in plant productivity over recent decades based on three global products. Global Biogeochem Cy, 34, e2020GB006613. https://doi.org/10.1029/2020GB006613, 2020.

Xie, X., Li, A., Jin, H., Bian, J., Zhang, Z., & Nan, X.: Comparing Three Remotely Sensed Approaches for Simulating Gross Primary Productivity over Mountainous Watersheds: A Case Study in the Wanglang National Nature Reserve, China. Remote Sensing, 13(18), 3567, https://doi.org/10.3390/rs13183567, 2021.

This a is an overall good manuscript, well written. The interest and complementary information of microwave data with respect to optical data for vegetation studies, in particular for the estimation of the Gross Primary Production is clear.

However, I have a number of concerns and questions for the authors.

Response: Dear Referee,

Thank you very much for reviewing our manuscript and your overall positive feedback. We are happy to answer your questions and we believe that we can dispel your concerns.

This GPP product comes from a VOD archive that it is produced as a "sub-product" of the ESA soil moisture Climate Change Initiative data set. In the CCI data set dense forest regions as masked because retrievals are not considered to be reliable, in particular those of AMSR-E, AMSR-2. Soil moisture and VOD are retrieved simultaneously, therefore why the retrievals are considered to be good for VODCA over those regions but not for SM CCI ?

Thank you for this important remark and question. First of all, we want to mention that although soil moisture and VOD are retrieved simultaneously, the retrieval algorithm distinguishes between emitted radiance coming from the soil surface and that coming from vegetation. Thus, VOD and SM, although retrieved simultaneously, can and should be regarded as separate products. It is correct that CCI soil moisture data is masked over densely vegetated regions. This is done because in these regions the largest part of microwave emission is caused by vegetation and thus almost the entire signal (i.e. the measured brightness temperature) comes from the vegetation, making soil moisture retrievals unreliable, but not VOD retrievals. Yet, VOD tends to saturate for very dense vegetation making it less likely to distinguish variability. We also observe this tendency for saturation in VODCA2GPP in our analysis (e.g., Fig A6). Nevertheless, our landcover-based analysis of mean yearly GPP (Fig. A3) suggests that VODCA2GPP compares very well with FLUXNET in-situ GPP in densely vegetated regions (Pearson's r: 0.89 for Evergreen Broadleaf Forest) which led us to the conclusion that VODCA2GPP is reliable also in regions with dense vegetation.

*This is indeed an important point, thus we added the following paragraph to the manuscript:*
Page: 19, Line: 440: *(…) Furthermore, VOD retrievals exhibit a tendency for saturation in regions with very dense vegetation making it less likely to distinguish variability. A slight tendency for saturation was also observed for VODCA2GPP but the landcover based analysis exhibited a very high agreement between VODCA2GPP and in-situ GPP indicating high reliability of VODCA2GPP over densely vegetated regions. (…)*

VOD depends on frequency. What is the physical meaning of a rescaling by CDF matching? It is not rescaling apples (VOD at C band, for isntance) and oranges (VOD at K band, for instance)? Can the

authors justify this approach and evaluate the impacts of this assumption into applications such as GPP estimation?

[Figure]

**Figure 2a: Temporal correlation between X-band- and C-band VODCA (left) and between X-band and Ku-band VODCA (right). Only spatially and temporally collocated data has been used. The correlations are based on the overlapping observation period (2002-01 – 2018-08).**

[Figure]

**Figure 2b: Temporal correlation between MODIS LAI and X-band VODCA (left) and between MODIS LAI and VODCA CXKu (right). Only spatially and temporally collocated data has been used. The correlations are based on the overlapping observation period (2002-01 – 2018-08).**

Thank you for this critical question. Indeed, VOD depends on the observed wavelength domain. Low-frequency observations such as those from L-band are sensitive to the water content in the whole vegetation, including the woody components, while high-frequency observations, such as C-, X-, and Ku-band, are more sensitive to the water content of the upper canopy layer (Li et al., 2021).

VODCA CXKu incorporates only high-frequency VOD products, namely C-, X-, and Ku-band, all of which indicate upper canopy dynamics and are highly correlated with each other. This is shown in Figure 2a, where for all biomes but those with little inter- and intra-annual variability (deserts and humid tropics) correlations are very high. In this figure, we show the correlation of C- (left) and Ku-band (right) with X-band VODCA, which has been used as scaling reference in the CDF-matching procedure. More so, in Figure 2b, we look at the agreement with MODIS LAI, which is an independent vegetation dataset related to leaf biomass (Tian et al., 2018). We can observe that the Spearman's R of MODIS LAI with X-band (left) and VODCA CXKu (right) are almost identical in all regions. This indicates that VODCA CXKu is very similar to the product used as scaling reference during CDF-matching, which is X-band.

However, VODCA CXKu exceeds the temporal length of the three single-frequency products, covering over 30 years of observation (1987 - 2020) and exhibits lower random error levels due to the merging approach employed. These features have led to an improved VODCA2GPP.

Line 138: ERA-5 Land resolution is not 8 km but 9 km

Thank you for making us aware of this. We revised this.

(Page: 5, Line 138): *(…) ERA5-Land is produced at a spatial resolution of 9 km (…)*

Line 187: I reckon that the dependency on time should be explicit in this equation or at least that the time scales at which those different VOD quantities are estimated should be explicit. Since VOD is approximately mdn(VOD)+delta(VOD) what is the real interest of adding a third term on VOD?

Thank you for this comment. The time scale at which delta(VOD) is derived is indeed crucial. We already incorporated this information in chapter 3.3 ("Preprocessing") but we agree that this term should already be explained at its first occurrence and in more detail. Thus, we added the following text (below Eq. 3.3) and removed the, now redundant, information from chapter 3.3:

(Page: 7, Line: 191): *Δ(VOD) is derived for each pixel ($x_i$) by computing the difference between two consecutive VOD observations of the smoothed and 8-daily aggregated VOD Signal (Teubner et al. 2019):*

$$\Delta(VOD) = VOD_{x_i,t_j} - VOD_{x_i,t_{j-1}}$$

*The smoothing was performed in order to increase the robustness of the derivation and implemented using a Savitzky-Golay filter with a window size of 11 data points as suggested by Teubner et al. (2021).*

Thank you for the question regarding the VOD term. For the answer to this question, we would primarily like to refer you to Teubner et al. (2019) who provide a detailed derivation of the theoretical background of the VOD2GPP model where also the relationship between GPP and VOD, delta(VOD), and mdn(VOD) is discussed in detail. Here, we provide a summary of the VOD2GPP theory which is based on Teubner et al. (2019):

For deriving the relationship between VOD and GPP we start with the relationship between GPP and ecosystem net uptake of carbon ($NPP$) and autotrophic respiration ($R_a$) (Bonan, 2008):

$GPP = R_a + NPP,$ (1)

The VOD2GPP-model is essentially based on the assumption that $R_\alpha$ can be expressed as differential equation (Ryan, 1990):

$R_\alpha = a_0 \left( \frac{dB}{dt} \right) + b_0\, B,$ (2)

The terms $\frac{dB}{dt}$ and $B$ denote biomass ($B$) and temporal changes in biomass ($\frac{dB}{dt}$) and they are proportional to the two constituents of $R_\alpha$, growth and maintenance respiration, respectively.

NPP can be approximately written as:

$NPP \approx \left( \frac{dB}{dt} \right) + loss\ terms,$ (3)

As the $loss\ terms$ only make a small fraction of NPP and are not directly reflected in VOD, they are neglected in the VOD2GPP-model (Teubner et al. 2019). By combining Eq. 1-3, we can express GPP via the following differential equation:

$$GPP = NPP + R_\alpha \approx a\left(\frac{dB}{dt}\right) + b\,B, \tag{4}$$

Another assumption is that AGB can be expressed as a function of VOD:

$$AGB = f(VOD) = \widehat{VOD} \tag{5}$$

Assuming that Biomass $B$ can be expressed as $AGB$, we can rewrite Eq. 4 and find the theoretical relationship between GPP and VOD:

$$GPP = a\left(\frac{d\widehat{VOD}}{dt}\right) + b\,\widehat{VOD} + c \tag{6}$$

Eq. 2 shows that temporal changes in VOD ($\sim \frac{dB}{dt}$) are needed to represent growth respiration and NPP while the bulk VOD signal ($\sim B$) is needed for representing the maintenance part of $R_\alpha$. As $R_\alpha$ exhibits a high sensitivity to temperature (Ryan et al., 1997) we included 2m surface temperature in an interaction term with VOD (Teubner et al., 2021). $Mdn(VOD)$ on the other hand corresponds approximately to the time-invariant offset c (Eq. 6) which is a static term and aids to convert VOD to GPP if the offset is not already represented in $VOD$. In other words, mdn(VOD) helps to make the VOD2GPP model more closely related to photosynthetically active parts of the vegetation by subtracting larger structural vegetation components (e.g., stems) which is also visible in Fig 1 c.

[Figure]

**Fig 1: Partial dependency plot for GPP and the input variables: VOD (a), $\Delta(VOD)$ (b), mdn(VOD) (c), T2M (d). Dashed lines denote the 95% confidence interval. The interaction term between VOD and T2M is depicted as 3D surface which is projected bin-wise onto a 2D plane for visualization.**

Figure 4: it is atypical to show the reference data in the y-axis. It is confusing for the reader. I strongly suggest inverting the axis. In addition, the linear regression equation should be shown or at least the slope and the intercept should be given in addition to R, RMSE and bias.

Thank you very much for making us aware of this. Having the reference data in the y-axis is indeed counter intuitive which is why we adapted our plots accordingly:

We swapped the x/y-axis in each scatter plot where the reference data (i.e., FLUXNET GPP) was in the y-axis (Fig. 4, Fig. A1, Fig. A2, Fig. A3). We added the linear regression line and equation in Fig. 4 and Fig. A2:

[Figure]

**Figure 4: GPP from FLUXNET plotted against GPP from VODCA2GPP, MODIS and FLUXCOM for the period 2002-2016 with 8-daily sampling.**

**References**

Bonan, G.: Ecological climatology: Concepts and applications, 2ed., Cambridge University Press, Cambridge, UK, New York, 550 pp., 2008.

Li, X., Wigneron, J. P., Frappart, F., Fan, L., Ciais, P., Fensholt, R., ... & Moisy, C. (2021). Global-scale assessment and inter-comparison of recently developed/reprocessed microwave satellite vegetation optical depth products. Remote Sensing of Environment, 253, 112208.

Ryan, M. G.: Growth and maintenance respiration in stems of Pinus contorta and Picea engelmannii, Can. J. Forest Res., 20, 48–57, doi:10.1139/x90-008, 1990.

Ryan, M. G., Lavigne, M. B., and Gower, S. T.: Annual carbon cost of autotrophic respiration in boreal forest ecosystems in relation to species and climate, J. Geophys. Res.-Atmos., 102, 28871–28883, 1997.

Teubner, I. E., Forkel, M., Camps-Valls, G., Jung, M., Miralles, D. G., Tramontana, G., van der Schalie, R., Vreugdenhil, M., Mösinger, L., and Dorigo, W. A.: A carbon sink-driven approach to estimate gross primary production from microwave satellite observations, Remote Sens. Environ., 229, 100–113, https://doi.org/10.1016/j.rse.2019.04.022, 2019.730

Teubner, I. E., Forkel, M., Wild, B., Mösinger, L., and Dorigo, W.: Impact of temperature and water availability on microwave-derived gross primary production, Biogeosciences, 18, 3285–3308, https://doi.org/10.5194/bg-18-3285-2021, 2021.

Tian, F., Wigneron, J.-P., Ciais, P., Chave, J., Ogée, J., Peñuelas, J., Ræbild, A., Domec, J.-C., Tong, X., Brandt, M., Mialon, A., Rodriguez-Fernandez, N., Tagesson, T., Al-Yaari, A., Kerr, Y., Chen, C., Myneni, R. B., Zhang, W., Ardö, J., and Fensholt, R.: Coupling of ecosystem-scale plant water storage and leaf phenology observed by satellite, Nat. Ecol. Evol., 2, 1428–1435, https://doi.org/10.1038/s41559-018-0630-3, 2018.